# Value-Based Deep RL Scales Predictably

**Oleh Rybkin** [1]  **Michal Nauman** [1 2]  **Preston Fu** [1]  **Charlie Snell** [1]  **Pieter Abbeel** [1]  **Sergey Levine** [1]  **Aviral Kumar** [3]

## Abstract

**Abstract:** Scaling data and compute is critical to the success of modern ML. However, scaling demands *predictability*: we want methods to not only perform well with more compute or data, but also have their performance be predictable from small-scale runs, without running the large-scale experiment. In this paper, we show that value-based off-policy RL methods are predictable despite community lore regarding their pathological behavior. First, we show that data and compute requirements to attain a given performance level lie on a *Pareto frontier*, controlled by the updates-to-data (UTD) ratio. By estimating this frontier, we can predict this data requirement when given more compute, and this compute requirement when given more data. Second, we determine the optimal allocation of a total resource *budget* across data and compute for a given performance and use it to determine hyperparameters that maximize performance for a given budget. Third, this scaling is enabled by first estimating predictable relationships between *hyperparameters*, which is used to manage effects of overfitting and plasticity loss unique to RL. We validate our approach using three algorithms: SAC, BRO, and PQL on DeepMind Control, OpenAI gym, and IsaacGym, when extrapolating to higher levels of data, compute, budget, or performance.

## 1. Introduction

Many latest advances in various areas of machine learning have emerged from training big models on large datasets. In this scaling guided research landscape, successfully executing even one single training run often requires a large amount of data, computational resources, and wall-clock time, such as weeks or months (Achiam et al., 2023; Team et al., 2023; Ramesh et al., 2022; Brooks et al., 2024). To maximize the success of these large-scale runs, the trend in the machine learning (ML) community has shifted toward not just performant, but also more **predictable algorithms that scale reliably** with more computation and training data size, such that downstream performance can be predicted from small-scale experiments, without actually running the large-scale experiment (McCandlish et al., 2018; Kaplan et al., 2020; Hoffmann et al., 2023; Dubey et al., 2024).

In this paper, we study if deep reinforcement learning (RL) is also amenable to such scaling and predictability benefits. We focus on **value-based methods** that train value functions using temporal difference (TD) learning, which are known to be performant at small scales (Mnih et al., 2015; Lillicrap et al., 2016; Haarnoja et al., 2018). Compared to policy gradient (Mnih et al., 2016; Schulman et al., 2017) and search methods (Silver et al., 2016), value-based RL can learn from arbitrary data and require less sampling or search, which can be inefficient or infeasible for open-world problems where environment interaction is costly.

We study scaling properties by predicting relationships between different **resources required for training**. *Data* requirement $\mathcal{D}$ is the amount of data needed to attain a certain level of performance. Likewise, *compute* requirement $\mathcal{C}$ refers to the amount of FLOPs or gradient steps needed to attain a certain level of performance. In RL uniquely, performance can be improved by increasing *either* available data or compute (e.g., training multiple times on the same data), which we capture via a *budget* requirement that combines data and compute $\mathcal{F} = \mathcal{C} + \delta \cdot \mathcal{D}$, where $\delta$ is some constant. An additive budget function is useful when the cost of data and compute can be expressed in similar units, such as wall-clock time or required finances.

To establish scaling relationships, we first require a way to predict the **best hyperparameter settings** at each scale. We find that learning rate $\eta$, batch size $B$, and the updates-to-data (UTD) ratio $\sigma$ are the most crucial hyperparameters for value-based RL. While supervised learning benefits from abundant theory to establish optimal hyperparameters (Krizhevsky, 2014; McCandlish et al., 2018; Yang et al., 2021), value-based RL often does not satisfy assumptions typical of supervised learning. For example, value-based RL needs to account for the non-i.i.d. nature of training data. Distribution shift due to periodic changes in the data col-

---

[1]UC Berkeley [2]University of Warsaw [3]CMU. Correspondence to: Oleh Rybkin <oleh.rybkin@gmail.com>, Aviral Kumar <aviralku@andrew.cmu.edu>.

*Proceedings of the $42^{nd}$ International Conference on Machine Learning*, Vancouver, Canada. PMLR 267, 2025. Copyright 2025 by the author(s).

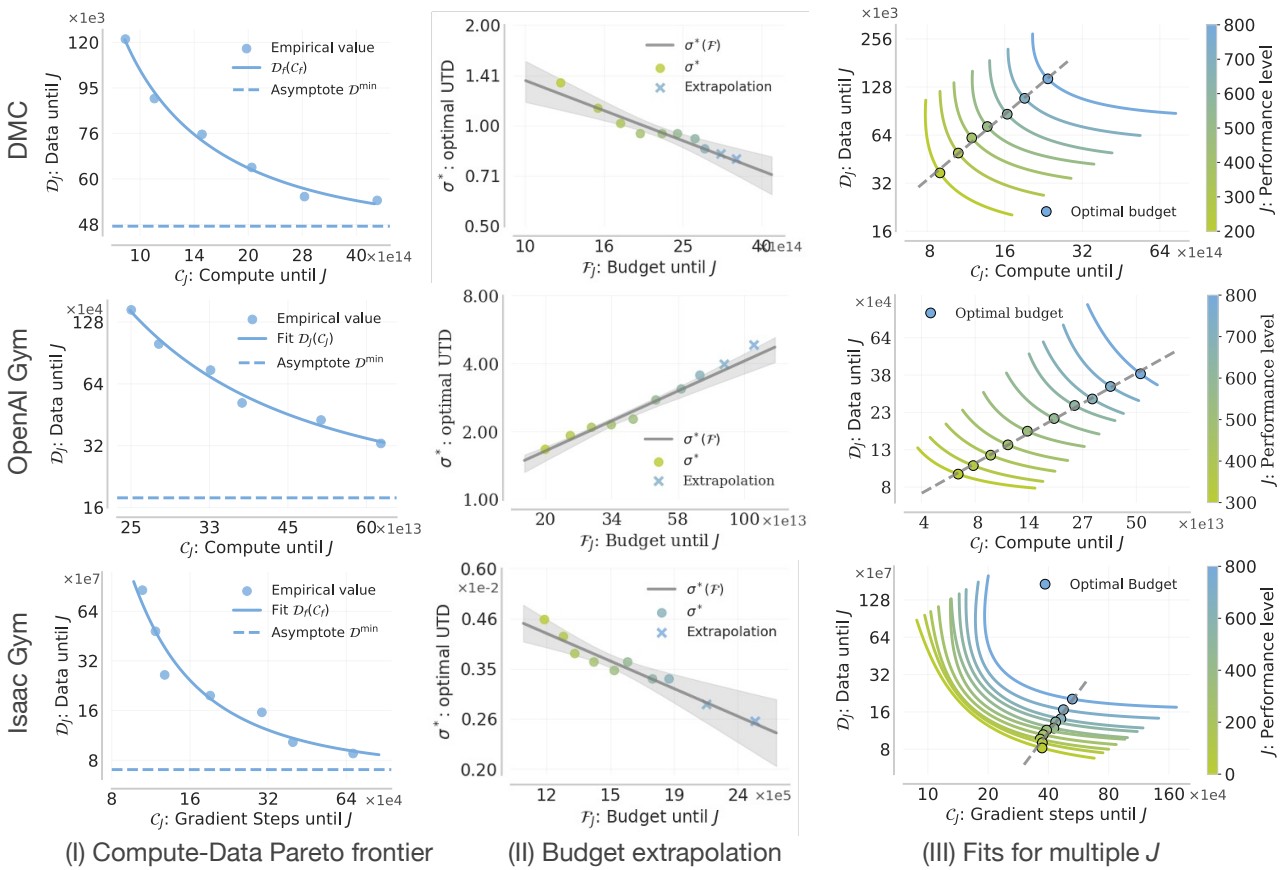

Figure 1: ***Scaling properties when increasing compute $\mathcal{C}$, data $\mathcal{D}$, budget $\mathcal{F}$, or performance $J$.* Left:** Compute versus data requirements Pareto frontier controlled by the UTD ratio $\sigma$. We observe that we can trade off data for compute and vice versa, and this relationship is predictable. **Middle:** Extrapolation from low to high performance. We observe that the optimal resource allocation controlled by $\sigma$ evolves predictably with increasing budget, and can be used to extrapolate from low to high performance. **Right:** Pareto frontiers for several performance levels $J$.

lection policy (Levine et al., 2020) contributes to a form of overfitting where minimizing training TD error may not result in a low TD error under the data distribution induced by the new policy. In addition, objective shift due to changing target values (Dabney et al., 2021) contributes to "plasticity loss" (D'Oro et al., 2023; Kumar et al., 2021). We show that it is possible to account for the training dynamics unique to value-based RL, and are able to find the best hyperparameters by setting the batch size and learning rate inversely proportional to the UTD ratio. We estimate this dependency using a power law (Kaplan et al., 2020), and observe that this model makes effective predictions.

Using the best predicted hyperparameters, we are now able to establish that data and compute requirements evolve as a predictable function of the UTD ratio $\sigma$. Furthermore, $\sigma$ defines the **tradeoff between data and compute**, which can be visualized as a Pareto frontier (Figure 1, left). Using this model, we are able to extrapolate the resource requirements from low-compute to high-compute setting, as well as from low-data to high-data setting as shown in the figure.

Using the Pareto frontiers, we are now able to **extrapolate**

**from low to high performance levels**. Instead of extrapolating as a function of return, which can be arbitrary and non-smooth, we extrapolate as a function of the allowed budget $\mathcal{F}$. We can define an optimal tradeoff between data and compute, and we observe that such optimal tradeoff value evolves predictably to higher budgets, which also attains a higher performance level (Figure 1, middle). Thus we are able to predict optimal hyperparameters, as well as data and compute allocation, for high-budget runs using only data from low-budget runs.

**Our contribution** is showing that the behavior of value-based deep RL methods based on TD-learning is predictable in larger data and compute regimes. Specifically, we:

1. establish predictable rules for dependencies between hyperparameters batch size ($B$), learning rate ($\eta$), and UTD ratio ($\sigma$) in value-based RL, and show that these rules enable more effective scaling.
2. show that data and compute required to attain a given performance level lie on a Pareto frontier, and are respectively predictable in the higher-compute or higher-data regimes.

3. show the optimal allocation of budget between data and compute, and predict how such allocation evolves with higher budgets for best performance.

Our findings apply to algorithms such as SAC, BRO, and PQL, and domains such as the DeepMind Control Suite (DMC), OpenAI Gym, and IsaacGym. The generality of our conclusions challenges conventional wisdom that value-based deep RL does not scale predictably.

## 2. RL Preliminaries and Notation

We study standard off-policy online RL, which maximizes the agent's return by training on a replay buffer and periodically collecting new data (Sutton and Barto, 2018). Value-based deep RL methods train a Q-network, $Q_\theta$, to minimize the temporal difference (TD) error:

$$L(\theta) = \mathbb{E}_\mathcal{P} \left[ \left( r(s,a) + \gamma \bar{Q}(s',a') - Q_\theta(s,a) \right)^2 \right], \quad (2.1)$$

where $\mathcal{P}$ is the replay buffer, $\bar{Q}$ is the target Q-network, $s$ denotes a state, and $a'$ is an action drawn from a policy $\pi(\cdot|s)$ that aims to maximize $Q_\theta(s,a)$. We implement this operation by sampling a batch of size $B$ from the buffer and taking a gradient step along the gradient of this loss with a learning rate $\eta$. In theory, off-policy algorithms can be made very sample efficient by minimizing the TD error fully over any data batch, which in practice translates to making more update steps to the Q-network per environment step, or higher "updates-to-data" ratio (UTD) (Chen et al., 2020). However, increasing the UTD ratio naïvely can lead to worse performance (Nikishin et al., 2022; Janner et al., 2019). To this end, unlike the standard supervised learning or LLM literature that considers $B$ and $\eta$ as two main hyperparameters affecting training (Kaplan et al., 2020; Hoffmann et al., 2023), our setting presents another hyperparameter, the UTD ratio $\sigma$, that we also study in our paper.

**Notation.** In this paper, we focus on the following key hyperparameters: the UTD ratio $\sigma$, learning rate $\eta$, and the batch size $B$. We will answer questions pertaining to performance of a policy $\pi$ denoted by $J(\pi)$, the total data utilized by an algorithm to reach a given target level of performance $J$ (denoted by $\mathcal{D}_J$), and the total compute budget utilized by the algorithm to reach performance $J$ (denoted by $\mathcal{C}_J$), which is measured in terms of FLOPs or wall-clock time taken by the algorithm.

## 3. Problem Statement and Formulation

To demonstrate that the behavior of value-based RL can be predicted reliably at scale, we first post multiple *resource optimization* questions that guide our scaling study. Viewing data and compute as two resources, we answer questions of the form: *what is the minimum value of [resource] needed to attain a given target performance? And what should the hyperparameters (e.g., $B, \eta, \sigma$) be in such this*

*training run?* We will answer such questions by fitting empirical laws from low data and compute runs to determine relationships between hyperparameters. Doing so, in turn, enables us to determine how to set hyperparameters and allocate resources to maximize performance when provided with a larger data and compute budget. Note that we wish to make these hyperparameter predictions without running the large data and compute budget experiment. While questions of this form have been studied in supervised learning, the answers are different for online RL, because online RL continuously collects its own data, which ties data and compute in a complex manner and breaks i.i.d. nature of datapoints.

Concretely, we study three resource optimization questions: **(1)** maximizing sample efficiency (i.e., minimize the amount of data $\mathcal{D}$ to attain a target performance under a given compute budget), **(2)** conversely, minimizing compute $\mathcal{C}$ (e.g., FLOPs or gradient steps, whichever is more appropriate) to attain a given performance given an upper bound on data that can be collected, and **(3)** maximizing performance given a total bound on data and compute.

---

**Problem 3.1** (Resource optimization problems). Find the best configuration $(B, \eta, \sigma)$ for algorithm Alg that minimizes either the data $\mathcal{D}$ or compute $\mathcal{C}$ consumed to obtain performance $J_0$:

1. *Maximal sample efficiency*:
$$(B^*, \eta^*, \sigma^*) := \arg \min_{(B,\eta,\sigma)} \quad \mathcal{D}$$
$$\text{s.t. } J\left(\pi_{\text{Alg}}(B,\eta,\sigma)\right) \geq J_0$$
$$\mathcal{C} \leq \mathcal{C}_0.$$

2. *Maximal compute efficiency*:
$$(B^*, \eta^*, \sigma^*) := \arg \min_{(B,\eta,\sigma)} \quad \mathcal{C}$$
$$\text{s.t. } J\left(\pi_{\text{Alg}}(B,\eta,\sigma)\right) \geq J_0$$
$$\mathcal{D} \leq D_0.$$

---

We solve these problems by fitting empirical models of the minimum data and compute needed to attain a target performance for different values of $J_0$. Doing so allows us to then solve the third setting **(3)** for maximizing performance given a total budget on data and compute as shown below.

---

**Problem 3.2** (Maximize performance at large data and compute budget). Find the best configuration $(B, \eta, \sigma)$ and resource allocations for data $\mathcal{D}$ and compute $\mathcal{C}$ that enable Alg to maximize performance at budget $\mathcal{F}_0$
$$(B^*, \eta^*, \sigma^*) := \arg \max_{(B,\eta,\sigma)} \quad J\left(\pi_{\text{Alg}}(B,\eta,\sigma)\right)$$
$$\text{s.t. } \mathcal{C} + \delta \cdot \mathcal{D} \leq \mathcal{F}_0.$$

---

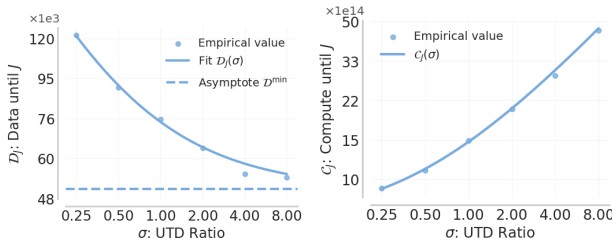

Figure 2: The data-compute tradeoff on DMC. *Left:* The minimum required data $\mathcal{D}_J$ scales with the UTD $\sigma$ as a power law. *Right:* The minimum required compute $\mathcal{C}_J$ increases with the UTD $\sigma$ as a sum of two power laws.

# 4. Scaling Results For Value-Based Deep RL

We will now present our main results addressing Problem 3.1 under the two settings discussed above. We will then use these results to present results for Problem 3.2. In order to do so, we run several experiments and estimate scaling trends from the results. Although this procedure might appear standard from scaling studies in language modeling, we found that instantiating it for value-based RL requires understanding the interaction of the various hyperparameters appearing in TD updates, and the data and compute efficiency of the algorithm. We will formalize these relationships via empirically estimated *laws* and show that these laws extrapolate reliably to new settings not used to obtain these empirical laws. Therefore, in this section, we present empirical and conceptual arguments to build functional forms of relationships between different hyperparameters. Before doing so, we provide our answers to Problems 3.1 and 3.2.

## 4.1. Main Scaling Results

We begin by answering Problem 3.1 where we maximize sample efficiency. We wish to estimate the minimal amount of data $\mathcal{D}_J$ needed to attain a given target performance, given an upper bound on compute $\mathcal{C} \leq \mathcal{C}_0$. To do so, we fit $\mathcal{D}_J$ needed to attain the target performance $J = J_0$ parameterized by the UTD ratio $\sigma$ (Eq. (4.1)). Intuitively, the minimum amount of data needed to attain a given performance is lower as more updates are made per datapoint (i.e., when $\sigma$ is high), as more "value" could be derived from the same datapoint. In addition, we would expect that even for the best value of $\sigma$, there is a minimum number of datapoints $\mathcal{D}^{\min}$ that are needed to learn given the "intrinsic" difficulty of the task at hand. Based on these intuitions, we hypothesize a power law relationship between $\mathcal{D}_J(\sigma)$ and $\sigma$, with an offset $\mathcal{D}^{\min}$ and constants $\alpha_J$ and $\beta_J$.

$$\mathcal{D}_J(\sigma) \approx \mathcal{D}_J^{\min} + \left(\frac{\beta_J}{\sigma}\right)^{\alpha_J} \qquad (4.1)$$

Empirical fits of $\mathcal{D}_J$ and $\sigma$ on the DMC suite are in Figure 2 and they validate the efficacy of this fit. We also emphasize that the existence of this power law makes $\mathcal{D}_J$ *predictable*, in that we can predict $\mathcal{D}_J$ for larger values of $\sigma$ that fall outside the range of $\sigma$ values used to get the fit (Figure 6).

---

**Scaling Observation 1: Data Requirements**

The amount of data $\mathcal{D}_J$ needed to reach a given return target $J_0$ decreases as a predictable function of the UTD $\sigma$, and is a power law (Eq. (4.1)).

To answer the optimization questions in Problem 3.1, we also need an expression for required compute until the target return $\mathcal{C}_J$. As $\sigma$ determines the number of gradient steps per data point, $\mathcal{C}_J$ is a function of $\sigma$. In particular, total compute is equal to the number of gradient steps taken multiplied by the parameter count of the model. Our study does not optimize over the model size and treats it as a constant. Thus, we write the compute $\mathcal{C}_J$ as a function of $\sigma$ as:

$$\mathcal{C}_J(\sigma) \approx 10 \cdot N \cdot B(\sigma) \cdot \sigma \cdot \mathcal{D}_J(\sigma) \qquad (4.2)$$

where $N$ denotes the model size, $B(\sigma)$ denotes the "best choice" batch size for a given UTD value $\sigma$, and other variables follow definitions from before. Note the additional factor of 10 in Eq. (4.2) emerges from the use of multiple forward passes to compute the loss function for value-based RL and the backward pass, through the Q-network (to contrast with language modeling, the typical multiplier is 6; the gap in our setting comes from the use of multiple forward passes). We plot $\mathcal{C}_J(\sigma)$ for different values of $\sigma$ and $J = J_0$ in Figure 2. Since $\mathcal{D}_J(\sigma)$ is not a constant and depends itself on $\sigma$, we note that this particular relationship between $\mathcal{C}_J(\sigma)$ and $\sigma$ is not a simple power law unlike Eq. (4.1). Instead, our derivation in Eq. (A.4) shows that $\mathcal{C}_J(\sigma)$ is given by a sum of two different power laws in $\sigma$. Similarly to $\mathcal{D}_J$, we also observe that the compute utilized is a *predictable* function of $\sigma$: we are able to accurately estimate the compute at larger values of $\sigma$ using the relationship in Eq. (4.2).

**Scaling Observation 2: Compute Requirements**

The compute $\mathcal{C}_J$ to attain a given return target $J_0$ increases as a predictable function of the UTD ratio $\sigma$, and is a sum of two power laws (Eq. (4.2)).

We observe that both required compute and data are controlled by the UTD ratio $\sigma$, which allows us to define a tradeoff between compute and data controlled by $\sigma$. We plot this tradeoff as a curve with compute $\mathcal{C}_J(\sigma)$ as $x$-axis and $\mathcal{D}_J(\sigma)$ as $y$-axis in Figure 1 (left). Further, as $\mathcal{D}_J(\sigma)$ is a monotonically decreasing function of $\sigma$, this curve defines a Pareto frontier: we can move left on the curve to increase data efficiency as the expense of compute and move right to increase compute efficiency at the expense of data. Also interestingly, due to the compute law being a sum of two power laws, in many environments there is a minimum $\sigma$ after which compute efficiency no longer improves as seen on OAI Gym in Figure 1.

**Solving for maximal data efficiency (Problem 3.1, (1)).** We can now solve Problem 3.1 in setting **(1)**. our strategy

to address setting **(1)** is to find the largest $\sigma$ (say $\sigma_{\max}$) that satisfies the compute constraint $C_J(\sigma) \leq C_0$, and then plug this $\sigma_{\max}$ into $D_J(\sigma)$ to obtain the data estimate. This approach enables us to express $D_J$ directly as a function of the available compute $C_0$, as we calculate in Eq. (4.2). This can be visualized as finding the value $D_J$ corresponding to some value $C_0$ on the Pareto frontier (Figure 1, left)

**Solving for maximal compute efficiency (Problem 3.1, (2)).** Likewise, the solution in **(2)** can be obtained by finding the smallest value of $\sigma$ in the range that satisfies the data constraint $D_J(\sigma) \leq D_0$, and computing the corresponding value of $C_J(\sigma)$. This can similarly be visualized on the Pareto frontier (Figure 1, left). We summarize our observations in terms of the following takeaway.

---

**Solving 3.1: The Compute-Data Pareto frontier**

The UTD ratio $\sigma$ defines a Pareto frontier between data and compute requirements, and estimating this frontier yields predictable solutions to resource optimization problems in settings **(1)** and **(2)**. Theoretically, the optimal $D_J^*$ for an available compute budget $C_0$ is:

$$D_J^*(C_0) \approx C_0 \cdot (10 \cdot N \cdot B(\sigma^*) \cdot \sigma^*)^{-1}. \quad (4.3)$$

The optimal $C_J$ for a given data budget $D_0$ is:

$$C_J^*(D_0) \approx 10 \cdot N \cdot B(\sigma^*) \cdot \sigma^* \cdot D_0. \quad (4.4)$$

Above, $\sigma^*$ denotes the minimizing UTD value. Calculation details are in Appendix A.

---

**Maximize return within a budget (Problem 3.2).** Finally, we tackle Problem 3.2 in order to extrapolate from low to high return. Here, we do not want to minimize resources, but rather want to maximize performance within a given total "budget" on data and compute. As discussed in Section 3, we consider budget functions linear in both data and compute, i.e., $\mathcal{F} = C + \delta \cdot D$, for a given constant $\delta$. Our estimated Pareto frontier in Eq. (4.4) will enable answering this question. To do so, we turn to directly predicting a good UTD value $\sigma^*$. This UTD value is one that not only leads to maximal performance, but also stays within the total resource budget $\mathcal{F}_0$. Once the UTD value has been identified, it prescribes a concrete way to partition the total resource budget into good data and compute requirements using the solutions to Problem 3.1.

We plot the data-compute Pareto frontiers for multiple values of $J_0$ in Figure 3 and in Figure 1 (right), and find that these curves move diagonally to the top-right for larger $J_0$. Intersecting these curves with iso-budget frontiers over $D$ and $C$ prescribed by the budget function, gives us the largest possible $J_0$ for which there is still a $(D, C)$ pair that just falls just within the budget $\mathcal{F}_0$ but attains performance $J_0$ (see Figure 3 for a worked out version of this procedure). Since both $D$ and $C$ are explained by $\sigma$, we can associate

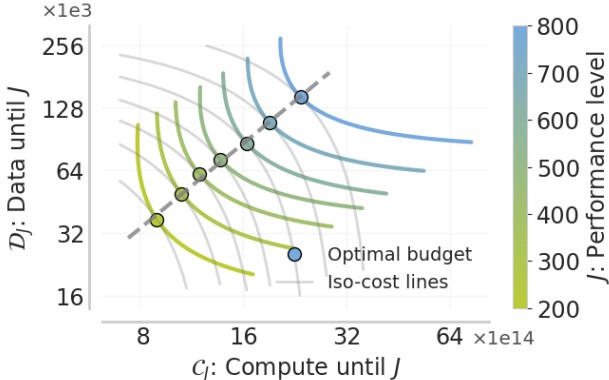

Figure 3: Visualization of the solution to Problem 3.2. Several Pareto frontiers (Figure 1, left) are shown, together with lines of iso-budget $\mathcal{F}$, which define optimal budget points $(D^*, C^*)$. Corresponding optimal UTD ratios $\sigma^*$ are a predictable function of the budgets $\mathcal{F}_0$, trend line shown dashed.

this point with a given $\sigma$ value. Hence, we can estimate the best value of $\sigma^*(\mathcal{F}_0)$ for a given budget threshold $\mathcal{F}_0$. Concretely, we observe a power law between $\sigma(\mathcal{F}_0)$ and $\mathcal{F}_0$, with constants $\beta_\sigma$ and $\alpha_\sigma$.

$$\sigma^*(\mathcal{F}_0) \approx \left( \frac{\beta_\sigma}{\mathcal{F}_0} \right)^{\alpha_\sigma}. \quad (4.5)$$

---

**Solving 3.2: Maximize Return Given a Budget**

The best UTD value $\sigma$ that leads to maximal $J$ is a predictable function of the budget $\mathcal{F}_0$ over data and compute, this relationship follows a power law, and also extrapolates to large budgets.

---

This relationship produces the optimal $\sigma$, and as a result, the optimal data and compute allocations to reliably attain maximum performance. As shown in Figure 1, estimating this law from low-budget experiments is sufficient for predicting good $\sigma$ values for large budget runs. These predicted $\sigma^*(\mathcal{F}_0)$ values extrapolate reliably to budgets outside the range used to fit this law (as shown by $\times$ in Figure 1). This concludes an exposition of our main results.

### 4.2. Fitting Relationships Between $(B, \eta, \sigma)$

To arrive at these scaling law fits above, we had to set hyperparameters $B$ and $\eta$, which we empirically observed to be important. We fit these hyperparameters as a function of $\sigma$, the only variable appearing in many of the scaling relationships discussed above. In this section, we will now describe how to estimate good values of $B$ and $\eta$ in terms of $\sigma$. Our analysis here relies crucially on the behavior of TD-learning that is distinct from supervised learning, where the UTD ratio $\sigma$ does not exist.

To understand relationships between batch size $B$, learning rate $\eta$, and the UTD ratio $\sigma$, we ran an extensive grid search. We first attempted to explain the relationship between the $B$

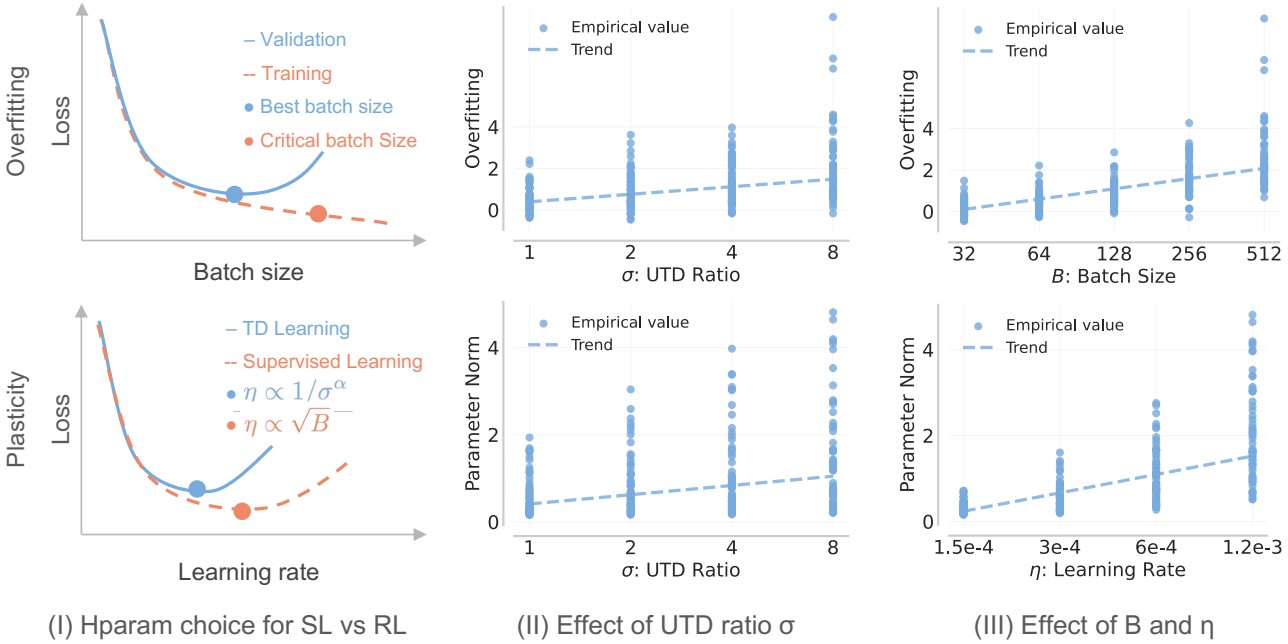

Figure 4: Hyperparameter effects in supervised learning and TD learning on DMC. ***Top:*** Overfitting increases with UTD while batch size can be used to counteract it. ***Bottom:*** Higher UTD leads to poor training dynamics and plasticity loss (D'Oro et al., 2023). Lower learning rates can be used to counteract it. While these relationships are not perfectly predictable, we use them to inform our design choices.

and $\eta$ values that attain the highest data efficiency (denoted $B^*, \eta^*$) using the standard heuristic in supervised learning: *when the batch size is smaller than the critical batch size, $B$ and $\eta$ are inversely correlated with each other* (McCandlish et al., 2018). However, as shown in Figure 5 (right), we find that without including the UTD ratio $\sigma$, best $B^*$ and $\eta^*$ exhibit very weak correlation. Further, the critical batch size (McCandlish et al., 2018) does not correlate with empirically best batch size as we show in Appendix F. Instead, surprisingly, we observe a strong correlation between $B^*$ and $\sigma$, as well as $\eta^*$ and $\sigma$, respectively. Since $B^*$ and $\eta^*$ exhibit near zero correlation among themselves, we can simply omit their dependency and opt for modeling them independently as a function of the UTD ratio, $\sigma$. We conceptually explain relationships between $B^*$ and $\sigma$, and $\eta^*$ and $\sigma$ below and show that models developed from this understanding enable us to reliably predict good values of $B$ and $\eta$, allowing us to fully answer Problem 3.1.

**Predicting best choice of $B$ in terms of $\sigma$.** Our proposed functional form for the best batch size $B^*$ takes the form of a power law in $\sigma$, which we also empirically validate in Figure 5 (left). We posit this form because, intuitively, large batch sizes increase the risk of overfitting because they lead to repetitive training on a fixed set of data. Furthermore, a small training loss on the distribution of data in the buffer does not necessarily reflect the behavior policy distribution of a learning agent (Levine et al., 2020). This means that minimizing the training loss to a large extent can result in poor test performance $J(\pi)$, as also seen by prior work (Li

et al., 2023a; Nauman et al., 2024a). One way to counteract this form of "overfitting" from a high UTD value $\sigma$ is to instead reduce the batch size in the run so that the training process sees a given sample fewer times. In fact, for a fixed UTD value $\sigma$, we empirically validate this hypothesis that a lower $B$ leads to substantially reduced overfitting on several tasks in Figure 4. Hence, we post an inverse relationship between the best batch size $B^*$ and the UTD value $\sigma$. We show in Figure 5 that indeed this inverse relationship can be estimated well by a power law, given formally as:

$$B^*(\sigma) \approx \left( \frac{\beta_B}{\sigma} \right)^{\alpha_B}. \qquad (4.6)$$

**Predicting best choice of learning rate $\eta$ as a function of $\sigma$.** Next we turn to understanding the relationship between $\eta$ and $\sigma$. We start from a simple observation: a very large $\sigma$ typically leads to worse performance not only due to overfitting but also due to plasticity loss (Kumar et al., 2021; D'Oro et al., 2023; Lyle et al., 2023), defined broadly as the inability of the value network to fit TD targets appearing later in training. Prior work states that plasticity loss is inherently related to the number of gradient steps performed and claims that larger norms of parameters of the Q-network are indicative of plasticity loss (D'Oro et al., 2023; Lyle et al., 2023). We would expect a larger learning rate to make higher magnitude updates against the same TD target, and hence move parameters to a state that suffers from difficulty in fitting subsequent targets (Dabney et al., 2021; Lee et al., 2024). As shown in Figure 4, the parameter norm indeed increases with a high learning rate. Therefore, given a UTD

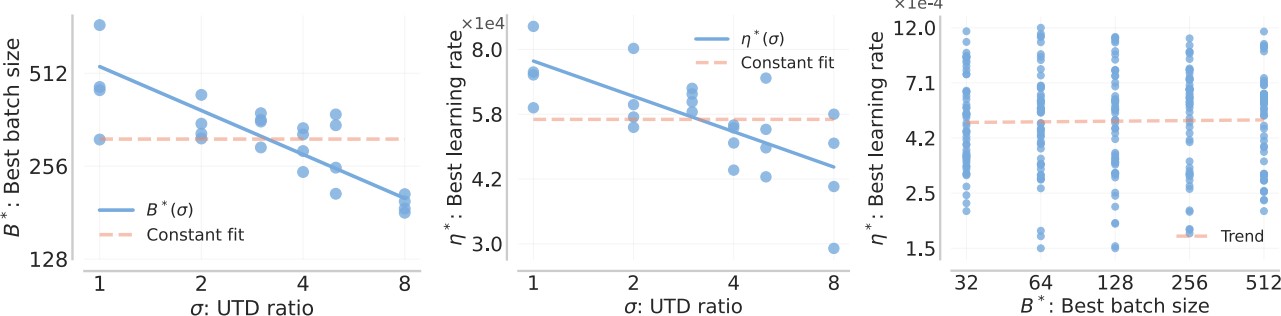

Figure 5: *Left, middle*: Fitting the best learning rate $\eta^*$ and batch size $B^*$ given UTD $\sigma$ on DMC. Modeling the dependency on $\sigma$ is crucial to obtain good hyperparameters, whereas using constant $B, \eta$ as is commonly done leads too poor extrapolation. *Right*: the best learning rate and batch size are not significantly correlated, a major difference from supervised learning.

value $\sigma$, we hypothesize that the best choice of learning rate, $\eta^*(\sigma)$ for a given performance should scale inversely in $\sigma$. Empirically we observe that this is indeed the case (Figure 5 (middle)), and we model this relationship:

$$\eta^*(\sigma) \approx \left(\frac{\beta_\eta}{\sigma}\right)^{\alpha_\eta}. \qquad (4.7)$$

> **Scaling Observation 3: Hyperparameter Selection**
>
> The best choices for the batch size and learning rate are predictable functions of the UTD $\sigma$, and both of these relationships follow a power law.

### 4.3. Empirical Workflow

> **Fitting Empirical Relationships**
>
> 1. Run a sweep for batch size $B$ and learning rate $\eta$ for several values of UTD $\sigma$. Since the batch size and learning rate are independent for the best $\sigma$, we can run these sweeps independently.
>
> 2. Estimate empirically the best of batch size $\tilde{B}$ and learning rate $\tilde{\eta}$, with statistical bootstrapping.
>
> 3. Fit $B^*(\sigma)$ and $\eta^*(\sigma)$ on $\tilde{B}, \tilde{\eta}$ according to Equations (4.6) and (4.7).
>
> 4. Using the found fits $B^*(\sigma), \eta^*(\sigma)$, run different values of $\sigma$ that cover a range spanning an order of magnitude; we use $16\times$, i.e., $\sigma_{\max}/\sigma_{\min} > 16$.
>
> 5. Fit $\mathcal{D}_J(\sigma)$ according to Eq. (4.1).
>
> 6. Using fits of $\mathcal{D}_J(\sigma)$ for different values of $J_0$, fit $\sigma^*(\mathcal{F}_0)$ according to Eq. (4.5).
>
> 7. Optimal hyperparameters can now be extrapolated to larger data, larger compute, or larger budget settings according to Problem 3.1.

Having presented solutions to Problems 3.1 and 3.2, we now present the workflow we utilize to estimate these empirical fits. Further details are in Section 5 and Appendix D. This

workflow can serve as a useful skeleton for scaling law studies with other value-based algorithms as well.

### 4.4. Evaluating Extrapolation

**Evaluating budget extrapolation**. Results on all environments are shown in Figure 1 (middle). We estimate several Pareto frontiers corresponding to points with equal changes in budget. We perform the $\sigma^*(\mathcal{F}_0)$ fit, while holding out two largest budgets. The quality of our fit for these two extrapolated budgets can be seen in the figure.

**Evaluating Pareto frontier extrapolation**. Results on OpenAI Gym are shown in Figure 6. We fit the data efficiency equation $\mathcal{D}_J(\sigma)$ Eq. (4.1) while holding out either two UTD values $\sigma$ with largest data requirement (left) or two $\sigma$ values with largest compute requirement (right). The quality of our fit for these two extrapolated $\sigma$ values can be seen in the figure.

**Hyperparameter fit extrapolation**. Results on OpenAI Gym are shown in Figure 6 (right). We plot the data efficiency fit when using hyperparameters according to our found dependency $B^*(\sigma), \eta^*(\sigma)$ (shown in red). These fits are estimated from $\sigma = 1, \ldots, 8$ and extrapolated to $\sigma = 0.5$. We compare the typical approach of tuning hyperparameters in online RL, where hyperparameters are tuned for one setting of $\sigma = 2$ and this setting is used for all UTD values (shown in blue). We see that our proposed hyperparameter fits improve results for values other than $\sigma = 2$. Further, this improvement is larger for larger values of $\sigma$, showing that accounting for hyperparameter dependency is critical.

## 5. Experimental Details

**Experimental Setup**  We focus on 12 tasks from 3 domains in our study. On **OpenAI Gym** (Brockman et al., 2016), we use Soft Actor Critic, a commonly used TD-learning algorithm (Haarnoja et al., 2018). We first run a sweep on 5 values of $\eta$, then a grid of runs with 4 values of $\sigma$ and 3 values of $B$, and then use hyperparameter fits

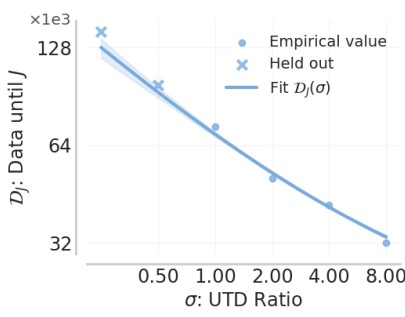 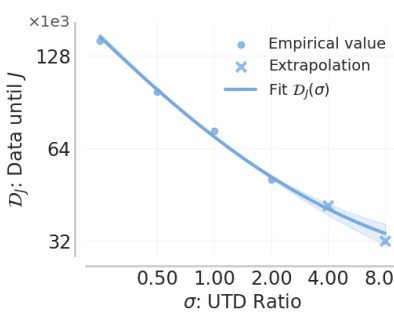 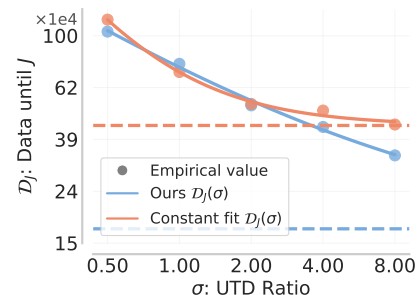

Figure 6: Extrapolation towards unseen values of $\sigma$ on OpenAI Gym. **Left:** We show Pareto frontier extrapolation towards higher data regime. **Middle:** We show Pareto frontier extrapolation towards higher compute regime. **Right:** We compare the best-performing hyperparameters (red) for $\sigma = 2$ to hyperparameters predicted via our proposed workflow (blue).

to run 2 more value of $\sigma$ with 8 seeds per task. To test our approach with larger models, we use **DMC** (Tassa et al., 2018), where, we utilize the state-of-the-art Bigger, Regularized, Optimistic (BRO) algorithm (Nauman et al., 2024b) that uses a larger and more modern architecture. We first run 5 values of $B$, 4 values of $\eta$, and 4 $\sigma$; and then use hyperparameters fits to run 2 more values of $\sigma$, with 10 seeds per task. Finally, we test our approach with more data on **IsaacGym** (Makoviychuk et al., 2021), where we use the Parallel Q-Learning (PQL) algorithm (Li et al., 2023b), which was designed to leverage massively parallel simulation like Isaac Gym that can quickly produce billions of environment samples. Because of computational expense, we only run one IsaacGym task. We first run 4 values of $\sigma$, 3 values of $\eta$, as well as 5 values of $B$, with 5 seeds per task, after which we run a second round of grid search with 7 values of $\sigma$. Further details are in Appendices B and D and Table 3.

**Fitting Functional Forms for Scaling Laws** We approximate Eq. (4.1) via brute-force search followed by LBFG-S with a log-MSE loss following (Hoffmann et al., 2023). For Equations (4.6) and (4.7), we fit a line in log space using least squares regression following Kaplan et al. (2020). In our experiments, we run a single fit that is shared across different tasks in a given benchmark. Specifically, we share the slope $\alpha_B, \alpha_\eta$ and use task-specific intercepts $\sigma_B^{\text{env}}, \sigma_\eta^{\text{env}}$ (as defined in Equations (4.6) and (4.7)) to be different for separate tasks. This technique is standard in ordinary least squares modeling and is referred to as fixed effect regression (Bishop and Nasrabadi, 2006). Sharing this slope serves the goal of variance reduction, which can be important if the granularity of the grid search over various hyperparameters run is coarse. More details are in Appendices B and D.

## 6. Related Work

**Scaling laws and predictability.** Prior work has studied scaling laws in the context of supervised learning (Kaplan et al., 2020; Hoffmann et al., 2023), primarily to predict

the effect of model size and training data on validation loss, while marginalizing out hyperparameters like batch size (McCandlish et al., 2018) and learning rate (Kaplan et al., 2020). There are several extensions of such scaling laws for language models, such as laws for settings with data repetition (Muennighoff et al., 2023) or mixture-of-experts (Ludziejewski et al., 2024), but most focus on cross-entropy loss, with an exception of Gadre et al. (2024), which focuses on downstream metrics. While scaling laws have guided supervised learning experiments, little work explores this for RL. The closest works are: Hilton et al. (2023) which fits power laws for on-policy RL methods using model size and the number of environment steps; Springenberg et al. (2024) who study model size scaling for offline RL; Jones (2021) which studies the scaling of AlphaZero on board games of increasing complexity; and Gao et al. (2023) which studies reward model overoptimization in RLHF. In contrast, we are the first ones to study predictability off-policy value-based RL methods that are trained via TD-learning. Not only do off-policy methods exhibit training dynamics distinct from supervised learning and on-policy methods (Kumar et al., 2022; Lyle et al., 2023), but we show that this distinction also results in a different functional form for scaling law altogether. We also note that while Hilton et al. (2023) use minimal compute, i.e., $\mathcal{C}_J$ in our notation as a metric of performance, our analysis goes further in several respects: **(1)** we also study the tradeoff between data and compute (Figure 1), **(2)** we can predict the algorithm configuration for best performance (Problem 3.1); **(3)** we study many budget functions ($\mathcal{C} + \delta \cdot \mathcal{D}$ can be any affine function).

**Methods for large-scale deep RL.** Recent work has scaled deep RL across three axes: model size (Kumar et al., 2023; Schwarzer et al., 2023; Nauman et al., 2024b), data (Kumar et al., 2023; Gallici et al., 2024; Singla et al., 2024), and UTD (Chen et al., 2020; D'Oro et al., 2023). Naïve scaling of model size or UTD often degrades performance or causes divergence (Nikishin et al., 2022; Schwarzer et al., 2023), mitigated by classification losses (Kumar et al., 2023), layer normalization (Nauman et al., 2024a), or feature normaliza-

tion (Kumar et al., 2022). In our work, we use scaled network architectures from Nauman et al. (2024b) (Section 5). In on-policy RL, prior works focus on effective learning from parallelized data streams in a simulator or a world model (Mnih et al., 2016; Silver et al., 2016; Schrittwieser et al., 2020). Follow-up works like IMPALA (Espeholt et al., 2018) and SAPG (Singla et al., 2024) use a centralized learner that collects experience from distributed workers with importance sampling updates. These works differ substantially from our study as we focus exclusively on value-based off-policy RL algorithms that use TD-learning and not on-policy methods. In value-based RL, prior work on data scaling focuses on offline (Yu et al., 2022; Kumar et al., 2023; Park et al., 2024) and multi-task RL (Hafner et al., 2023). In contrast, we study online RL and fit scaling laws to answer resource optimization questions.

## 7. Discussion, Limitations, and Future Work

In this paper, we show that value-based deep RL algorithms scale predictably. We establish relationships between good values of hyperparameters of value-based RL. We then establish a relationship between required data and required compute for a certain performance. Finally, this allows us to determine an optimal allocation of resources to either data and compute. Although only estimated from small-scale runs, our empirical models reliably *extrapolate* to large compute, data, budget, or performance regimes. Despite folk wisdom to the contrary, we show it is possible to predict behavior of value-based off-policy RL algorithms at larger scale using small-scale experiments.

At the same time, this first study also presents a number of open questions and challenges:

1. While simple power law models work well, an open question remains as to whether such laws are theoretically grounded, and whether there are better and more refined functional forms.
2. Our study only focused on three hyperparameters ($B$, $\eta$, and $\sigma$). We do not focus on optimal tradeoff between model size and UTD, which is important for compute scaling. For data efficient RL, it is important to analyze the dependency of weight decay and weight reset frequency on UTD, which are typical tricks employed by many of the most performant methods in literature.
3. While we focus on online RL, it is important to study scaling of offline-to-online and offline RL, which will allow direct applications of scaling law findings to large model training.
4. Finally, while we study relatively small models, future work will focus on verifying our results with larger model scales, larger scale tasks, study the effect of modern architectures, and cover a larger range of compute scales spanning multiple orders of magnitude.

Our work is only one step in studying scaling laws for value-based RL methods. Further research has the potential to improve our understanding of value-based RL at scale, provide researchers with tools to focus innovation on more important components, and eventually provide guidelines towards scaling value-based RL similarly to scaling enjoyed by other modern deep learning approaches.

## Acknowledgements

We would like to thank Zhang-Wei Hong, Amrith Setlur, Rishabh Agarwal, Seohong Park, and Max Simchowitz for feedback on an earlier version of this paper. We would like to thank Andrea Zanette, Seohong Park, Kyle Stachowicz, and Qiyang Li for informative discussions. This research was supported by ONR under N00014-24-12206, N00014-22-1-2773, and ONR DURIP grant, with compute support from the Berkeley Research Compute, Polish high-performance computing infrastructure, PLGrid (HPC Center: ACK Cyfronet AGH), that provided computational resources and support under grant no. PLG/2024/017817. Pieter Abbeel holds concurrent appointments as a Professor at UC Berkeley and as an Amazon Scholar. This work was done at UC Berkeley and CMU, and is not associated with Amazon.

## Impact Statement

This paper aims to contribute to the advancement of reinforcement learning. While our work may have various societal implications, none warrant specific emphasis here.

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

# Appendices

## A. Additional details on derivations

**FLOPs calculation.** Recall that FLOPs per forward and backward passes are equal to $\mathcal{C}_J^{\text{forward}}(\sigma) \approx 2 \cdot N \cdot B(\sigma) \cdot \sigma \cdot \mathcal{D}_J(\sigma)$ and $\mathcal{C}_J^{\text{backward}}(\sigma) \approx 4 \cdot N \cdot B(\sigma) \cdot \sigma \cdot \mathcal{D}_J(\sigma)$, with $\sigma$ denoting the number of gradient steps per environment steps. Q-learning methods used in our study use MLP and ResNet architectures, which are well modeled with this approximation. Assuming same size for actor and critic as an approximation, a training iteration of the critic requires three forward passes and one backward pass, totaling $\mathcal{C}_J^{\text{critic}}(\sigma) \approx 10 \cdot N \cdot B(\sigma) \cdot \sigma \cdot \mathcal{D}_J(\sigma)$. A training iteration of the actor requires two forward and two backward passes, totaling $\mathcal{C}_J^{\text{actor}}(\sigma) \approx 12 \cdot N \cdot B(\sigma) \cdot \sigma \cdot \mathcal{D}_J(\sigma)$. Here we follow the standard practice of updating the actor every time a new data point collected, while the critic is updated according to the UTD ratio $\sigma$. Since we expect the critic to be updated more then the actor. As such, in this study we assume

$$\mathcal{C}_J(\sigma) \approx \mathcal{C}_J^{\text{critic}}(\sigma) \approx 10 \cdot N \cdot B(\sigma) \cdot \sigma \cdot \mathcal{D}_J(\sigma). \tag{A.1}$$

**Compute and sample efficiency.** Following Eq. (4.1), the number of data points required to achieve performance $J$ is equal to:

$$\mathcal{D}_J(\sigma) \approx \mathcal{D}_J^{\min} + \left(\frac{\beta_J}{\sigma}\right)^{\alpha_J} \tag{A.2}$$

Given the expressions for required data points, practical batch size, and FLOPs Equations (4.1), (4.6) and (A.1), we can now derive the expression for compute required to reach a particular performance expressed in terms of $\sigma$. First, note that the number of parameter updates is

$$\sigma \cdot \mathcal{D}_J(\sigma) \approx \sigma \cdot \mathcal{D}_J^{\min} + \frac{\beta_J^{\alpha_J}}{\sigma^{\alpha_J - 1}} \tag{A.3}$$

Combining above, Eq. (4.6) with Eq. (A.1) yields:

$$\begin{aligned}
\mathcal{C}_J(\sigma) &\approx 10 \cdot N \cdot B(\sigma) \cdot \left(\sigma \cdot \mathcal{D}_J^{\min} + \frac{\beta_J^{\alpha_J}}{\sigma^{\alpha_J - 1}}\right) \\
&\approx 10 \cdot N \cdot \left(\frac{\beta_B}{\sigma}\right)^{\alpha_B} \cdot \left(\sigma \cdot \mathcal{D}_J^{\min} + \frac{\beta_J^{\alpha_J}}{\sigma^{\alpha_J - 1}}\right) \\
&\approx 10 \cdot N \cdot \left(\frac{\mathcal{D}_J^{\min} \cdot \beta_B^{\alpha_B}}{\sigma^{\alpha_B - 1}} + \frac{\beta_J^{\alpha_J} \cdot \beta_B^{\alpha_B}}{\sigma^{\alpha_J + \alpha_B - 1}}\right).
\end{aligned} \tag{A.4}$$

We observe that the resulting expression is a sum of two power laws. In practice, one of the power laws will dominate the expression and a simple mental model is that compute increases with UTD as a power law with a coefficient $< 1$ (see Figure 2).

**Maximal compute efficiency.** Here, we solve the compute optimization problem presented in Section 3. We write the problem:

$$(B^*, \eta^*, \sigma^*) := \arg\min_{(B, \eta, \sigma)} \mathcal{C} \quad \text{s.t.} \quad J\left(\pi_{\text{Alg}}(B, \eta, \sigma)\right) \geq J_0 \ \wedge \ \mathcal{D} \leq D_0. \tag{A.5}$$

Firstly, we formulate the Lagrangian $\mathcal{L}$:

$$\mathcal{L}(\sigma, \lambda) = \mathcal{C}_J(\sigma) + \lambda \cdot (\mathcal{D}_J(\sigma) - D_0)$$
$$\approx 10 \cdot N \cdot B(\sigma) \cdot \left(\sigma \cdot \mathcal{D}_J^{\min} + \frac{\beta_J^{\alpha_J}}{\sigma^{\alpha_J - 1}}\right) + \lambda \cdot \left(\mathcal{D}_J^{\min} + \left(\frac{\beta_J}{\sigma}\right)^{\alpha_J} - \mathcal{D}_0\right) \tag{A.6}$$

Here, the constrained with respect to performance $J_0$ is upheld through the use of $\mathcal{C}_J(\sigma)$ and $\mathcal{D}_J(\sigma)$ which are defined such that $J = J_0$. We proceed with calculating the derivative with respect to $\lambda$ to find the minimal $\sigma$ that is able to achieve the desired sample efficiency $\mathcal{D}_J$. We denote such optimal UTD as $\sigma^*$:

$$\frac{\partial \mathcal{L}}{\partial \lambda} = \mathcal{D}_J^{\min} + \left(\frac{\beta_J}{\sigma}\right)^{\alpha_J} - \mathcal{D}_0 = 0 \quad \Longrightarrow \quad \sigma^* = \frac{-\beta_J}{\left(\mathcal{D}_J^{\min} - \mathcal{D}_0\right)^{1/\alpha_J}} \tag{A.7}$$

Then, we substitute the $\sigma^*$ into the expression defining compute, as well as use Eq. (4.6):

$$\mathcal{C}_J(\sigma^*) \approx 10 \cdot N \cdot \frac{\beta_B^{\alpha_B}}{\sigma^{\alpha_B - 1}} \cdot \left(\mathcal{D}_J^{\min} + \frac{\beta_J^{\alpha_J}}{\sigma^{\alpha_J}}\right)$$
$$\approx 10 \cdot N \cdot \frac{\beta_B^{\alpha_B}}{(\sigma^*)^{\alpha_B - 1}} \cdot \left(\mathcal{D}_J^{\min} + \frac{\beta_J^{\alpha_J} \cdot \left(\mathcal{D}_J^{\min} - \mathcal{D}_0\right)}{-\beta_J^{\alpha_J}}\right) \tag{A.8}$$
$$\approx 10 \cdot N \cdot \beta_B^{\alpha_B} \cdot (\sigma^*)^{1 - \alpha_B} \cdot \mathcal{D}_0$$

**Maximal sample efficiency.** Firstly, we note that we treat $B(\sigma)$ as a constant and do not optimize with respect to it. We start with the problem definition:

$$(B^*, \eta^*, \sigma^*) := \arg \min_{(B, \eta, \sigma)} \quad \mathcal{D} \quad \text{s.t.} \quad J\left(\pi_{\text{Alg}}(B, \eta, \sigma)\right) \geq J_0 \quad \wedge \quad \mathcal{C} \leq C_0. \tag{A.9}$$

Similarly to the maximal compute efficiency problem, we formulate the Lagrangian $\mathcal{L}$:

$$\mathcal{L}(\sigma, \lambda) = \mathcal{D}_J(\sigma) + \lambda \cdot (\mathcal{C}_J(\sigma) - C_0)$$
$$\approx \mathcal{D}_J^{\min} + \left(\frac{\beta_J}{\sigma}\right)^{\alpha_J} + \lambda \cdot \left(10 \cdot N \cdot B(\sigma) \cdot \sigma \cdot \left(\mathcal{D}_J^{\min} + \frac{\beta_J^{\alpha_J}}{\sigma^{\alpha_J}}\right) - \mathcal{C}_0\right) \tag{A.10}$$

Again, we uphold the constraint with respect to the performance through the use of $\mathcal{D}_J(\sigma)$ and $\mathcal{C}_J(\sigma)$. We calculate the derivative with respect to $\lambda$:

$$\frac{\partial \mathcal{L}}{\partial \lambda} = 10 \cdot N \cdot B(\sigma) \cdot \sigma \cdot \left(\mathcal{D}_J^{\min} + \frac{\beta_J^{\alpha_J}}{\sigma^{\alpha_J}}\right) - \mathcal{C}_0 = 0 \quad \Longrightarrow \quad \mathcal{D}_J^{\min} + \frac{\beta_J^{\alpha_J}}{\sigma^{\alpha_J}} = \frac{\mathcal{C}_0}{10 \cdot N \cdot B(\sigma) \cdot \sigma} = \mathcal{D}_J \tag{A.11}$$

Since $\mathcal{D}_J$ is monotonic in $\sigma$ and does not model impact of $B$ on the sample efficiency, the optimization problem can be solved via Weierstrass extreme value theorem. As such, we find the biggest $\sigma$ and that fulfills the compute constraint, and find the data requirement for such $\sigma$.

## B. Experimental details

For our experiments, we use a total of 12 tasks from 3 benchmarks (DeepMind Control (Tunyasuvunakool et al., 2020), Isaac Gym (Makoviychuk et al., 2021), and OpenAI Gym (Brockman et al., 2016)). We list all considered tasks in Table 1.

Table 1: Tasks used in presented experiments.

| Domain | Task | Optimal $\pi$ Returns |
|---|---|---|
| DeepMind Control | Cartpole-Swingup | 1000 |
| | Cheetah-Run | 1000 |
| | Dog-Stand | 1000 |
| | Finger-Spin | 1000 |
| | Humanoid-Stand | 1000 |
| | Quadruped-Walk | 1000 |
| | Walker-Walk | 1000 |
| Isaac Gym | Franka-Push | 0.05 |
| OpenAI Gym | HalfCheetah-v4 | 8500 |
| | Walker2d-v4 | 4500 |
| | Ant-v4 | 6625 |
| | Humanoid-v4 | 6125 |

**Figure 1.** We use all available UTD values for the fits, which is 6 for DMC, 5 for OAI Gym, and 7 for Isaac Gym. Given the dependency of compute and data on UTD, we plot the resulting curve. We average the data efficiencies across all tasks in each domain, as described in Appendix D. For plots on the left, we use $J = 800$.

We calculate compute given the model sizes of $N = 4.92e6$ for DMC, $N = 1.5e5$ for OAI Gym, and $N = 2e6$ following standard implementations of the respective algorithms.

For budget extrapolation, we use tradeoff values $\delta$ to mimic the wall-clock time of the algorithm. We use $\delta = 1e10$ for DMC, $\delta = 5e9$ for OAI Gym, and $\delta = 1e4$ for Isaac Gym. We exclude runs affected by resets ($\sigma = 8$) for DMC since the returns right after the reset are lower, which adds noise to the results.

**Figure 2.** We use the same data as for DMC in Figure 1 (left).

**Figure 3.** We use the same data as for DMC in Figure 1 (right).

**Figure 4.** **Left**: we show an illustration that reflects our observed empirical results about the dependencies between hyperparameters.

**Right, middle**: we investigate the correlations between overfitting, parameter norm of the critic network, and $\sigma$. We observed the same relationships on all tasks. Here, to avoid clutter, we plot 3 tasks from DMC benchmark: cheetah-run, dog-stand, and quadruped-walk. To measure overfitting, we compare the TD loss calculated on samples randomly sampled from the buffer (corresponding to *training data*) to TD loss calculated on 16 newest transitions (corresponding to *validation data*) according to:

$$\text{Overfitting} = TD^{\text{training}} - TD^{\text{validation}}. \tag{B.1}$$

We fit the linear curves using ordinary least squares with mean absolute error loss.

**Figure 5.** In the left and central Figures, we evaluate the $B^*$ and $\eta^*$ models. For each DMC task, we find the best hyperparameters according to our workflow and procedure described in Section 5 and Appendix D. While the intercepts vary across environments, for simplicity we plot data points and fits from all environments in the same figure by shifting them with the corresponding intercept. In the right Figure, we marginalize over $\sigma$ and visualize best performing pairs of $B$ and $\eta$.

**Figure 6.** Here, we investigate 4 tasks from OpenAI Gym, listed in Table 1, and compare the extrapolation performance of two hyperparameter sets: the best performing hyperparameters for $\sigma = 1$, found by testing 8 different hyperparameter values listed in Table 3 (we refer to this configuration as *baseline*); and hyperparameters predicted by our proposed models of $B^*$ and $\eta^*$. We fit our models using $\sigma \in (1, 2, 4, 8)$, and extrapolate to $\sigma \in (0.5, 16)$. The graph shows the data efficiency with threshold as 700, normalized according to the procedure in Appendix D.

**Figure 7.** The goal of the left Figure is to visualize the effects of isotropic regression fit on a noisy data. We use the SciPy package (Virtanen et al., 2020) to run the isotropic model. In the right Figure we visualize the process of best hyperparameter selection using bootstrapped confidence intervals. We describe the bootstrapping strategy in Appendix D.

## C. Resulting Fits

**DMC**    Refer to Table 5 for environment-specific values.

$$
\begin{aligned}
\eta^* &= \beta_\eta \cdot \sigma^{-0.26} \\
B^* &= \beta_B \cdot \sigma^{-0.47} \\
\mathcal{D}_J &= \mathcal{D}^{\min} \cdot \left(1 + \left(\frac{\sigma}{0.45}\right)^{-0.74}\right) \\
\sigma^* &= 1.4\mathrm{e}8 \cdot \mathcal{F}_0^{-0.53}
\end{aligned}
\tag{C.1}
$$

**OpenAI Gym**    Refer to Table 5 for environment-specific values.

$$
\begin{aligned}
\eta^* &= \beta_\eta \sigma^{-0.30} \\
B^* &= \beta_B \sigma^{-0.33} \\
\mathcal{D}_J &= \mathcal{D}^{\min} \cdot \left(1 + \left(\frac{\sigma}{4.02}\right)^{-0.69}\right) \\
\sigma^* &= 1.4\mathrm{e}8 \cdot \mathcal{F}_0^{-0.53}
\end{aligned}
\tag{C.2}
$$

**Isaac Gym**

$$
\begin{aligned}
\eta^* &= 8.77 \cdot \left(1 + \left(\frac{\sigma}{2.57\mathrm{e}\text{-}3}\right)^{-0.26}\right) \\
B^* &= 38.6 \cdot \left(1 + \left(\frac{\sigma}{1.42\mathrm{e}\text{-}2}\right)^{-0.68}\right) \\
\mathcal{D}_J &= 6.8\mathrm{e}7 \cdot \left(1 + \left(\frac{\sigma}{1.88}\right)^{-0.87}\right) \\
\sigma^* &= 11.3 \cdot \mathcal{F}_0^{-0.57}
\end{aligned}
\tag{C.3}
$$

Table 2: Coefficients for DMC and OpenAI Gym fits.

| Domain | Task | $\beta_\eta$ | $\beta_B$ | $\mathcal{D}^{\min}$ |
|---|---|---|---|---|
| DMC | cartpole-swingup | 7.55e-4 | 538.2 | 2.4e4 |
| | cheetah-run | 6.25e-4 | 564.9 | 3.5e5 |
| | finger-spin | 8.77e-4 | 608.2 | 2.9e4 |
| | humanoid-stand | 3.86e-4 | 451.8 | 3.8e5 |
| | quadruped-walk | 8.46e-4 | 526.4 | 6.2e4 |
| | walker-walk | 9.38e-4 | 313.3 | 3.3e4 |
| OpenAI Gym | Ant-v4 | 1.35e-4 | 447.0 | 2.7e5 |
| | HalfCheetah-v4 | 1.86e-3 | 415.4 | 7.8e4 |
| | Humanoid-v4 | 1.65e-4 | 351.6 | 1.8e5 |
| | Walker2d-v4 | 7.85e-4 | 399.1 | 1.7e5 |

## D. Additional details on the fitting procedure

**Preprocessing return values.**    In order to estimate the fits from our laws, we need to track the data and compute needed by a run to hit a target performance level. Due to stochasticity both in training and and evaluation, naïve measurements of this

Table 3: Tested configurations.

| Hyperparameters | DeepMind Control | Isaac Gym | OpenAI Gym |
|---|---|---|---|
| Updates-to-data $\sigma$ | 1, 2, 4, 8 | $\frac{1}{1024}, \frac{1}{2048}, \frac{1}{4096}, \frac{1}{8192}, \frac{1}{16384}, \frac{1}{32768}, \frac{1}{65536}$ | 1, 2, 4, 8 |
| Batch size $B$ | 32, 64, 128, 256, 512 | 512, 1024, 2048, 4096, 8192 | 128, 256, 512 |
| Learning rate $\eta$ | 15e-5, 3e-4, 6e-4, 12e-3 | 1e-4, 2e-4, 3e-4 | 1e-4, 2e-4, 5e-4, 1e-3, 2e-3 |

point can exhibit high variance. This in turn would result in low-quality fits for $\mathcal{D}_J$ and $\mathcal{C}_J$. Thus, we preprocess the return values before estimating the fits by running isotonic regression (Barlow and Brunk, 1972). Isotonic regression transforms return values to the most aligned monotonic sequence of values that can then be used to estimate $\mathcal{D}_J$. While in general return values can decrease with more training after reaching a target value, and this will result in a large deviation between the isotonic fit and true return values, the proposed isotonic transformation still suffices for us as our goal is to simply fit the *minimum* number of samples or compute needed to attain a target return. As we can still make reliable predictions that extrapolate to larger scales, the downstream impact of this error is clearly not substantial. We also average across random seeds before running isotonic regression to further reduce noise. We normalize the returns for all environments to be between 0 and 1000 (Table 1 lists pre-normalized returns), and reserve the points of 700 and 800 for budget extrapolation in Figure 1.

**Uncertainty-adjusted optimal hyperparameters.** While averaging across seeds and applying isotonic regression reduces noise, we observe that the granularity of our grid search on learning rate and batch size limits the precision of the resulting hyperparameter fits $\tilde{B}, \tilde{\eta}$. Noise due to random seed generation makes hyperparameter selection harder as some hyperparameters that appear empirically optimal might simply be so due to noise. We observe that we can correct for this precision loss by constructing a more precise estimate of $\tilde{B}, \tilde{\eta}$ adjusted for this uncertainty. Specifically, we run $K = 100$ bootstrap estimates by sampling $n$ random seeds with replacement out of the original $n$ random seeds, applying isotonic regression, and selecting the optimal hyperparameters $\tilde{B}_k, \tilde{\eta}_k$. We then use the mean of this bootstrapped estimate to improve the precision:

$$\tilde{B}_{\text{bootstrap}} = \frac{1}{K} \sum_k \tilde{B}_k$$

$$\tilde{\eta}_{\text{bootstrap}} = \frac{1}{K} \sum_k \tilde{\eta}_k \tag{D.1}$$

We have also experimented with more precise laws for learning rate and batchsize by adding an additive offset. In this case, we follow Hoffmann et al. (2023) and fit the data using brute-force search followed by LBFG-S. We use MSE in log space as the error: $\text{MSE}_{\log}(a, b) = (\log a - \log b)^2$.

$$B^*(\sigma) \approx B_{\min} + \frac{\sigma_B}{\sigma^{\alpha_B}} \tag{D.2}$$

$$\eta^*(\sigma) \approx \eta_{\min} + \frac{\sigma_\eta}{\sigma^{\alpha_\eta}}. \tag{D.3}$$

However, we found that this more complex fit did not validate the decrease of degrees of freedom given a limited sweep range, resulting in accuracy of extrapolation.

**Independence of $B$ and $\eta$.** Whereas the optimal choice of $B$ and $\eta$ is often intertwined as UTD changes, we observe in our experiments that the correlation between them is relatively low (Figure 5). If we ran a cross-product grid search with hyperparameter space $\{B_1, \dots, B_{n_B}\} \times \{\eta_1, \dots, \eta_{n_\eta}\}$, we can use this fact to further improve the results by averaging the estimate $\tilde{B}$ over different values of $\eta$. That is, we produce the estimate $\tilde{B}^{[\eta=\eta_i]}$ (respectively $\tilde{\eta}^{[B=B_i]}$) by only looking at the runs where $\eta = \eta_i$, and averaging such estimates.

$$\tilde{B}_{\text{mean}} = \frac{1}{n_\eta} \sum_i \tilde{B}^{[\eta=\eta_i]}$$

$$\tilde{\eta}_{\text{mean}} = \frac{1}{n_B} \sum_i \tilde{\eta}^{[B=B_i]} \tag{D.4}$$

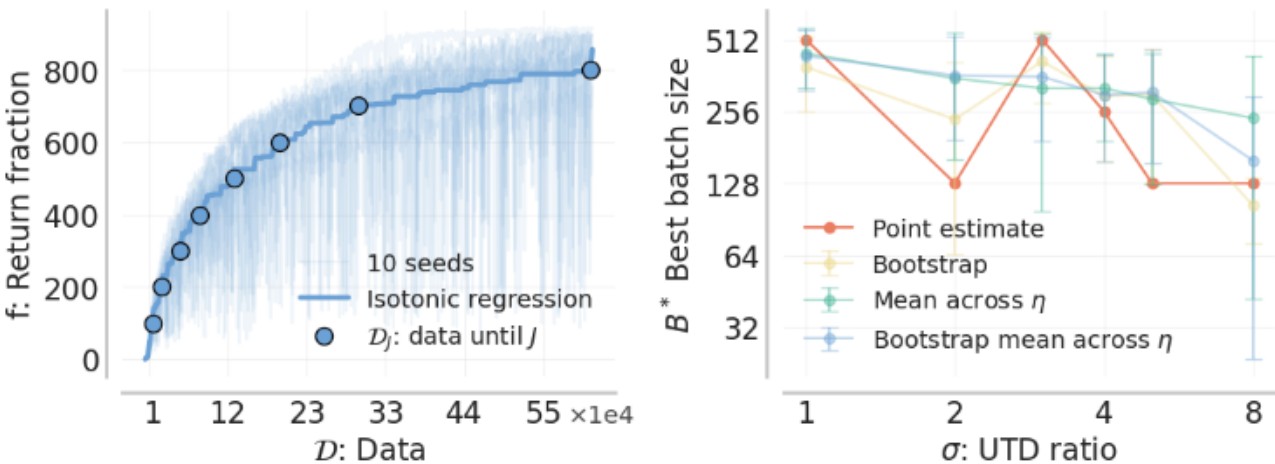

Figure 7: *Left:* Determining performance via isotonic regression on DMC. ***Right:*** improving hyperparameter selection with uncertainty adjustment on DMC. Further details are in Appendix D.

**Data efficiency.** We fit data efficiency of the runs with our found practical hyperparameters $B^*, \eta^*$ according to Eq. (4.1). We follow Hoffmann et al. (2023) and fit the data using brute-force search followed by LBFG-S. We use MSE in log space as the error: $\mathrm{MSE}_{\log}(a, b) = (\log a - \log b)^2$.

In DeepMind Control Suite, we would like to share the data efficiency fit across different environments env. We normalize the data efficiency $\mathcal{D}$ by the intra-environment median data efficiency medians $\mathcal{D}_{\mathrm{med}}^{\mathrm{env}} = \mathrm{median}\{\mathcal{D}_{[\sigma=\sigma_i]}^{\mathrm{env}}|i = 1..n_\sigma\}$. For interpretability, we further re-normalize $D$ with the overall median $\mathcal{D}_{\mathrm{med}}$: $\mathcal{D}_{\mathrm{norm}} = \mathcal{D} \cdot \mathcal{D}_{\mathrm{med}}/\mathcal{D}_{\mathrm{med}}^{\mathrm{env}}$. We will need to express the data efficiency law alternatively as:

$$D_J(\sigma) \approx \mathcal{D}_J^{\min}\left(1 + \left(\frac{\beta_J}{\sigma}\right)^{\alpha_J}\right). \tag{D.5}$$

This is equivalent to Eq. (4.1) because the coefficient $\beta_J$ absorbs $\mathcal{D}_J^{\min}$. However, this expression makes explicit an overall multiplicative offset[1] $\mathcal{D}_J^{\min}$. Our median normalization is then equivalent to fitting per-environment coefficients $\mathcal{D}_J^{\min}$, following our procedure for environment-shared hyperparameter fits. However, we further improve robustness by fixing the per-environment coefficients to be the median data efficiency and do not require fitting them.

## E. Additional experimental results

Table 4: Correlation coefficients for empirically optimal DMC hyperparameters.

|  | R |
| --- | --- |
| learning rate and batch size | 0.04 |
| batch size and UTD | -0.40 |
| learning rate and UTD | -0.46 |

Table 5: Error of Pareto frontier extrapolation.

|  | R |
| --- | --- |
| toward larger compute | 7.8% |
| toward larger data | 10.6% |

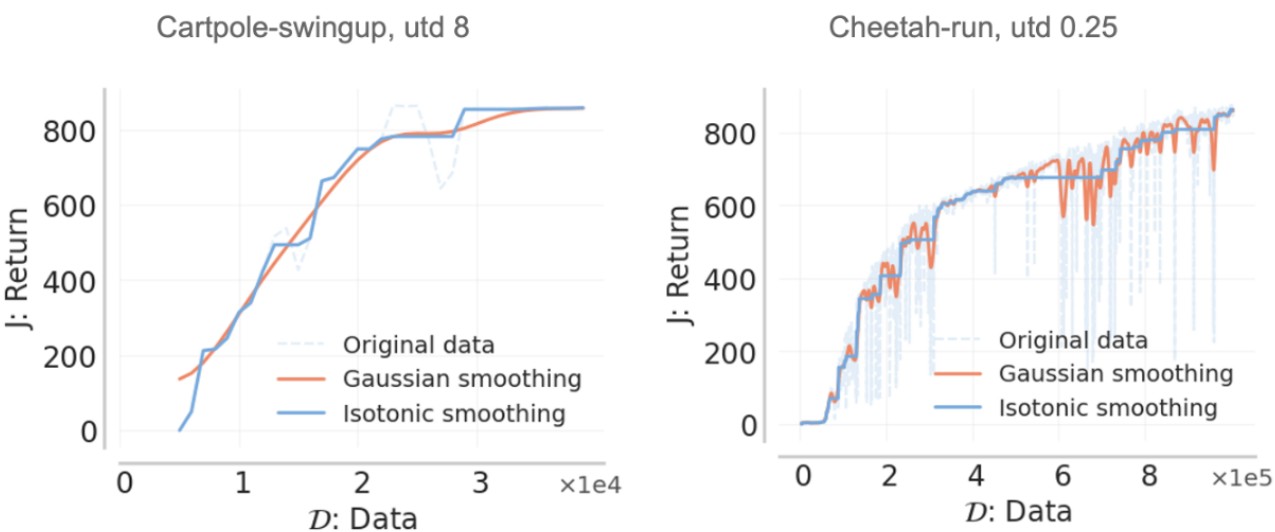

Figure 8: Another example of isotonic regression. Using gaussian smoothing with variance $\sigma = 3$ leads to both oversmoothing (right) and undersmoothing (left).

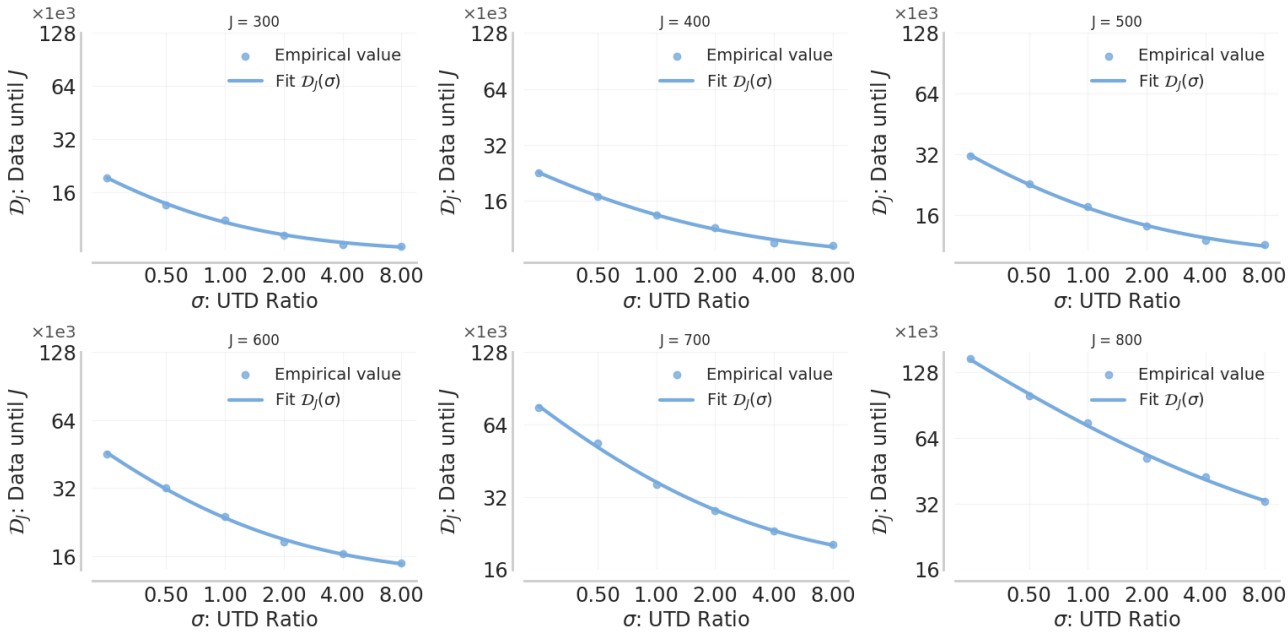

Figure 9: Additional fit results on OpenAI gym for different values of $J$.

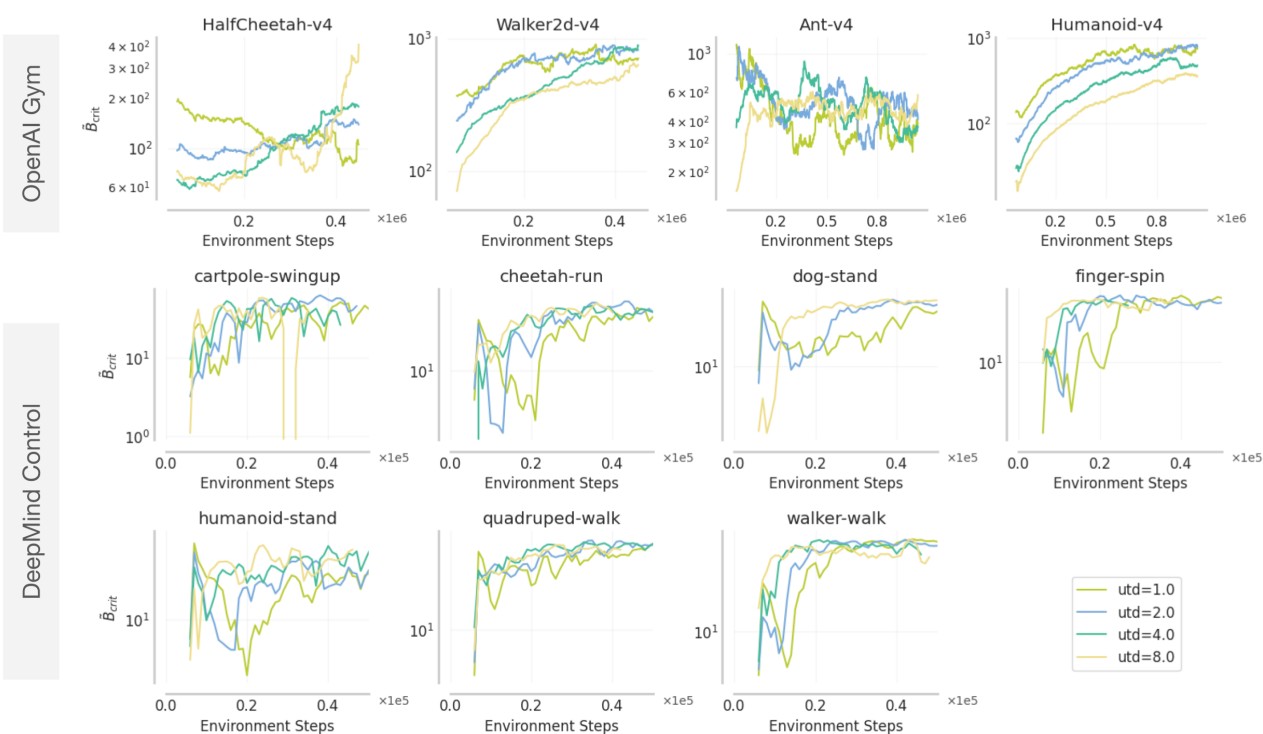

Figure 10: An approximation of the critical batch size over training. Further details are in Appendix F.

## F. Critical batch size analysis

Previous work has argued that there is a critical batch size $B_{\text{crit}}$ for neural network training in image classification, generative modeling, and reinforcement learning with policy gradient algorithms (McCandlish et al., 2018) — a transition point at which increasing the batch size begins to yield diminishing returns. We follow this work and compute an estimate of the gradient noise scale $B_{\text{noise}} \approx B_{\text{crit}}$ according to the following procedure: throughout training, we compute the gradient norm $|G_B|$ of the critic network for batches of size $B = B_{\text{small}} := 64$ and $B = B_{\text{big}} := 1024$. Then, we evaluate

$$|\mathcal{G}|^2 := \frac{1}{B_{\text{big}} - B_{\text{small}}} \left( B_{\text{big}} |G_{B_{\text{big}}}|^2 - B_{\text{small}} |G_{B_{\text{small}}}|^2 \right)$$

$$\mathcal{S} := \frac{1}{1/B_{\text{small}} - 1/B_{\text{big}}} \left( |G_{B_{\text{small}}}|^2 - |G_{B_{\text{big}}}|^2 \right)$$

and take $\tilde{B}_{\text{crit}} := \mathcal{S}/|\mathcal{G}|^2$. In practice, to account for the noisiness of $|G|^2$, we first take rolling averages of $|G_{B_{\text{small}}}|$ and $|G_{B_{\text{big}}}|$ over training, and tune the window size so that the estimates for $|\mathcal{G}|^2$ and $\mathcal{S}$ are stable.

We show the values of $\tilde{B}_{\text{crit}}$ over training in Figure 10. Unlike policy gradient methods, we find that the critical batch size (averaged over training) has little correlation with the optimal batch size, as shown in Figure 11.

---

[1]This form enforces that $\mathcal{D}_J^{\min}$ is positive.

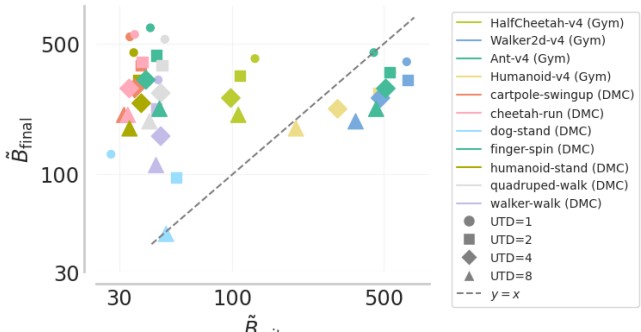

Figure 11: $\tilde{B}_{\text{final}}$ vs. $\tilde{B}_{\text{crit}}$, grouped by task and UTD.

Table 6: Batch size values predicted by the proposed model on DMC.

| Task | $\sigma = 0.25$ | $\sigma = 0.5$ | $\sigma = 1$ | $\sigma = 2$ | $\sigma = 4$ | $\sigma = 8$ |
|---|---|---|---|---|---|---|
| cartpole-swingup | 1040 | 752 | 544 | 384 | 288 | 208 |
| cheetah-run | 1088 | 784 | 560 | 400 | 288 | 208 |
| dog-stand | 240 | 176 | 128 | 96 | 64 | 48 |
| finger-spin | 1168 | 848 | 608 | 432 | 320 | 224 |
| humanoid-stand | 864 | 624 | 448 | 320 | 240 | 176 |
| quadruped-walk | 1008 | 736 | 528 | 384 | 272 | 192 |
| walker-walk | 608 | 432 | 320 | 224 | 160 | 112 |

Table 7: Learning rate values predicted by the proposed model on DMC.

| Task | $\sigma = 0.25$ | $\sigma = 0.5$ | $\sigma = 1$ | $\sigma = 2$ | $\sigma = 4$ | $\sigma = 8$ |
|---|---|---|---|---|---|---|
| cartpole-swingup | .00108 | .000902 | .000755 | .000631 | .000528 | .000442 |
| cheetah-run | .000893 | .000747 | .000625 | .000523 | .000438 | .000366 |
| dog-stand | .000664 | .000555 | .000465 | .000389 | .000325 | .000272 |
| finger-spin | .00125 | .00105 | .000877 | .000734 | .000614 | .000514 |
| humanoid-stand | .000551 | .000461 | .000386 | .000323 | .00027 | .000226 |
| quadruped-walk | .00121 | .00101 | .000846 | .000708 | .000592 | .000496 |
| walker-walk | .00134 | .00112 | .000938 | .000785 | .000657 | .000549 |

Table 8: Batch size values predicted by the proposed model on OpenAI Gym.

| Task | $\sigma = 0.25$ | $\sigma = 0.5$ | $\sigma = 1$ | $\sigma = 2$ | $\sigma = 4$ | $\sigma = 8$ | $\sigma = 16$ |
|---|---|---|---|---|---|---|---|
| Ant-v4 | 704 | 560 | 448 | 352 | 288 | 224 | 176 |
| HalfCheetah-v4 | 672 | 528 | 416 | 336 | 256 | 208 | 160 |
| Humanoid-v4 | 560 | 432 | 352 | 272 | 224 | 176 | 144 |
| Walker2d-v4 | 640 | 496 | 400 | 320 | 256 | 192 | 160 |

Table 9: Learning rate values predicted by the proposed model on OpenAI Gym.

| Task | $\sigma = 0.25$ | $\sigma = 0.5$ | $\sigma = 1$ | $\sigma = 2$ | $\sigma = 4$ | $\sigma = 8$ | $\sigma = 16$ |
|---|---|---|---|---|---|---|---|
| Ant-v4 | .000206 | .000167 | .000138 | .000109 | .000087 | .000070 | .000060 |
| HalfCheetah-v4 | .002820 | .002280 | .001900 | .001510 | .001210 | .000972 | .000827 |
| Humanoid-v4 | .000251 | .000203 | .000169 | .000134 | .000107 | .000086 | .000073 |
| Walker2d-v4 | .001180 | .000958 | .000806 | .000640 | .000512 | .000412 | .000347 |

Table 10: Batch size values predicted by the proposed model on IsaacGym.

| Task | $\sigma = \frac{1}{65536}$ | $\sigma = \frac{1}{32768}$ | $\sigma = \frac{1}{16384}$ | $\sigma = \frac{1}{8192}$ | $\sigma = \frac{1}{4096}$ | $\sigma = \frac{1}{2048}$ | $\sigma = \frac{1}{1024}$ |
|---|---|---|---|---|---|---|---|
| Franka-Push | 7927 | 5105 | 3234 | 2030 | 1269 | 791 | 493 |

Table 11: Learning rate values predicted by the proposed model on IsaacGym.

| Task | $\sigma = \frac{1}{65536}$ | $\sigma = \frac{1}{32768}$ | $\sigma = \frac{1}{16384}$ | $\sigma = \frac{1}{8192}$ | $\sigma = \frac{1}{4096}$ | $\sigma = \frac{1}{2048}$ | $\sigma = \frac{1}{1024}$ |
|---|---|---|---|---|---|---|---|
| Franka-Push | 0.000317 | 0.000265 | 0.000221 | 0.000185 | 0.000154 | 0.000129 | 0.000107 |

