# OpenReview forum: "Value-Based Deep RL Scales Predictably"
_ICML.cc/2025/Conference — ICML 2025 poster_

### Official Review · Reviewer_vVTn · 2025-02-20

**Overall Recommendation:** 3

**Summary:**

This paper studies the scaling law in value-based RL methods. In particular, it provides a thorough analysis on how different components, such as batch size and number of gradient steps, affect the performance and computation budget.

***
## update after rebuttal

I have read the rebuttal as well as other responses. I will keep my score.

**Claims And Evidence:**

Claims made in the paper are supported by clear and convincing evidence.

**Essential References Not Discussed:**

Essential references are adequately discussed.

**Experimental Designs Or Analyses:**

Extensive empirical results are presented.

**Methods And Evaluation Criteria:**

This paper has made extensive evaluation in both theoretical and empirical aspects.

**Other Comments Or Suggestions:**

Please see other parts of the review.

**Other Strengths And Weaknesses:**

+ I believe this work tries to handle a very important problem. Scalability is perhaps the biggest challenge in off-policy algorithms, and this study attempts to demonstrate that they can still be scaled effectively. The methodology used to identify the optimal hyperparameter settings is also interesting.

+ In practice, off-policy algorithms are considered to have poor scalability, not only due to the large batch sizes and excessive gradient steps but also because they require a buffer to store all the data collected so far, which leads to high memory usage. Additionally, the wall-time efficiency of off-policy methods is poor, as they often take much longer to train an agent compared to on-policy algorithms. Therefore, it would be more convincing if it could be demonstrated that value-based methods can scale in terms of both memory usage and wall-time efficiency.

**Questions For Authors:**

+ What is the wall-time efficiency of value-based algorithms in terms of scalability? For example, How long does the SAC algorithm take to complete a 5M environment-step training in DMC?

+ Does it improve the memory usage when using the optimal hyperparameter?

**Relation To Broader Scientific Literature:**

None.

**Theoretical Claims:**

There is no proof in the paper.

---

> ### Author Rebuttal · Authors · 2025-04-01
>
> Thank you for the feedback and a positive review of the paper. We are glad that you find our evidence clear and convincing. We answer your questions below. Please let us know if you find your concerns addressed, and if so we would be grateful if you would be willing to raise your score. We are happy to address any remaining concerns.
>
> > What is the wall-time efficiency of value-based algorithms in terms of scalability? For example, How long does the SAC algorithm take to complete a 5M environment-step training in DMC?
>
> The wall-clock efficiency of these algorithms is highly dependent on the implementation of both the agent and simulator, as well as the computing infrastructure. For an NVIDIA A100 GPU, UTD=1 configuration, and DeepMind Control Suite of tasks (which present a CPU-based simulator), training for 5M environment steps should take around 12 hours. In our study, we model compute by making it proportional to the product of the model size, training length, UTD, and batch size (equation 4.2 of our manuscript) akin to how we measure FLOPs for language model training. As such, our compute metric is highly correlated to actual wall clock requirements. To avoid confusion, we will add an explanation to our compute metric definition where we discuss its relation to wall clock requirements, as well as add wall clock graphs of our tested algorithms to the appendix. Thank you for the suggestion!
>
> > Does it improve the memory usage when using the optimal hyperparameter
>
> Our study uses a fixed memory size between all variants of algorithms (i.e. we use a fixed model and replay buffer sizes). As such, we left the study of memory/scaling interaction for future work. We hypothesize that the optimal or “good enough” hyperparameters indeed improve the memory usage of the algorithm, in a sense that it allows for training a successful agent with either less data or less parameters.
>
> Beyond the changes described above, we add a number of improvements suggested by other reviewers (e.g., additional quantitative analysis of proposed scaling curves, estimation of scaling laws for different levels of J), which we invite you to inspect under the following link: https://sites.google.com/view/value-based-rl-scales/ and have released the code here: https://colab.research.google.com/drive/1BaqvAMb6svGojAuiOV8qFAUrZQwfPlDg?usp=sharing. We hope that these changes increase the reviewers' confidence in our work. If so, we kindly ask to consider updating the score of our work.

---

> > ### Comment · Reviewer_vVTn · 2025-04-01
> >
> > Thanks for the response, it has addressed my concerns. My feedback remains positive.

---

### Official Review · Reviewer_sgc1 · 2025-02-20

**Overall Recommendation:** 1

**Summary:**

The main claim of the paper is that it is possible to predict optimal hyperparameters, data quantity and compute allocation for high budget from low-budget experiments. This is broken down into the following sub-claims:

1. The amount of data needed for a given performance is predictable as a function of the UTD according to power law defined in Equation 4.1.
2. The amount of data needed for a given performance is predictable as a function of the UTD according to power law defined in Equation 4.2.
3. For a given desired performance and budget, there exists a predictable “best” UTD that follows a power law in Equation 4.5. This relationship extrapolates to larger budgets.
4. The optimal choice of batch size and learning rate are predictable functions of the UTD.

**Claims And Evidence:**

**1. The amount of data needed for a given performance is predictable as a function of the UTD according to power law defined in Equation 4.2.**

Intuitively, this first claim is mostly believable. The literature contains many examples that show there are sample efficiency benefits to higher UTD ratios, with diminishing returns when properly regularized. To defend this claim, the authors present Figure 2 (left). I do not find this figure sufficient to defend the claim for the following reasons:
1. It is unclear what $J$ is.
2. It is unclear whether the result holds for all values of $J$.
3. It is unclear how accurate the result is outside of visual inspection.
4. The scale of the y-axis follows an unknown exponential scheme that makes visual inspection difficult.
5. It is unclear how the result was averaged over the collection of DMC environments.
6. It is unclear whether the result holds over individual environments, or only the average.

Furthermore, I’m not sure I buy that the result necessarily extends to some asymptote. For example, the challenges associated with offline RL and high UTDs have been well-documented [Kumar 2019, Levine 2020, Li 2023], requiring specific algorithmic modifications. If we were to push the UTD to some extreme values, we should expect to see either (1) collapse/divergence, where an algorithm never achieves a desired performance or (2) increasing regularization/algorithmic modifications (e.g., frequent resets, policy regularization) that would also impact the quantity of data required.

There also appears to be some missing experimental details. Knowing that most methods do not scale naively to higher UTD values [Chen 2021, Li 2023, D’Oro 2023], what modifications were used with the base algorithms to obtain these results?

---

**2. The amount of compute needed for a given performance is predictable as a function of the UTD according to a sum of power laws in Equation 4.2.**

Similarly, this claim is believable based on prior results in the literature. To defend this claim, the authors present Figure 2 (right). I do not find this claim to be adequately defended. On top of my concerns mentioned above for Claim 1,
- It is unclear how the compute is affected by the unknown variable quantity $B(\sigma)$ (“best choice” batch size for each UTD).

As an aside, it is unclear to me where the “sum” of power laws appears in Equation 4.2, as it consistently solely terms multiplied together.

---

**3. For a given desired performance and budget, there exists a predictable “best” UTD that follows a power law in Equation 4.5. This relationship extrapolates to larger budgets.**

To defend this claim, the authors reference Figure 3 and Figure 1 (right). Again, I do not find that these figures provide sufficient evidence to defend the authors’ claims.
- It is unclear how these results are generated. What is the process for determining data points? For example, the way I would expect a data point to be produced would be the following (this is my best guess, because the process is not documented).
    1. Pick an environment and a UTD value.
    2. Determine the best hyperparameters for this UTD value.
    3. Run the algorithm until some performance J is obtained.
    4. Report the (compute, data).

However, in this process, neither compute, nor data is a fixed quantity. How are empirical values averaged over a collection of environments?

This is highly exacerbated by the fact that different environments learn at very different rates. For example, for the OpenAI gym environments, the authors use SAC but only include 4 out of the 5 environments used by the original SAC paper (Table 1). Furthermore, the values of J seem arbitrarily selected. Based on the original SAC paper [Haarnoja, 2018], we can see that a score of 8500 on HalfCheetah is obtainable in fewer than 1M timesteps, but a score of 6625 on Ant would take more than 3M timesteps. Looking at more modern results, the authors of Cross-Q [Bhatt, 2024] report SAC results, where a score > 6000 is not obtained before 3M timesteps on Ant, but a score of 8500 on HalfCheetah is obtained in fewer than 1M timesteps.

- Figures 3 and 1 (right) do not directly show UTD. While each of these points corresponding to optimal budget do correspond to some value of UTD, we do not know what that UTD value is, and obtaining this UTD value has been obfuscated by an unknown “best choice” batch size that varies between UTD values. We cannot use this information to verify Equation 4.5, nor the claim that this power law extrapolates.

---

Continued in "other strengths and weaknesses" for Claim 4, due to a lack of space.

**Essential References Not Discussed:**

No issues.

**Experimental Designs Or Analyses:**

It is unclear whether the experimental design is valid because they are not adequately described. See my issues with the claims made by the paper, and the lack of scientific rigor in the presentation of the results.

**Methods And Evaluation Criteria:**

The evaluation criteria appears to be visual inspection. I do not find this sufficient.

I find the scope of benchmarks (3 domains) to be sufficient. I do think more than one algorithm per benchmark is necessary when attempting to prescribe sweeping statements.

**Other Comments Or Suggestions:**

N/A.

**Other Strengths And Weaknesses:**

Missing comments on Claim 4:

**4. The optimal choice of batch size and learning rate are predictable functions of the UTD.**

The authors defend this claim with Figure 4. Figure 4 suffers from the same lack of scientific rigour as many of the previous figures.

1. It is unclear how the best batch size or learning rate is determined over a collection of environments.
2. It is unclear if the results hold over different domains and algorithms.
3. It is unclear if the results hold over individual environments.
4. It is unclear how accurate the relationship is between UTD and best batch size/learning rate outside of visual inspection.
5. It is unclear if the results hold over different target performance levels or budgets.

While I do find the results are more plainly presented than previous figures, and the trends appear to be more clear, I am not confident that there is sufficient evidence to claim the existence of a widely-applicable power law.

---

I have additional concerns about the scientific rigor in Figure 1.
1. For the middle row (Budget extrapolation), there are an inconsistent number of points across environments.
2. Given this inconsistency, it’s unclear how the target performance J was determined for each point.
3. The exponential scale, and use of linear extrapolation makes the relationship appear linear at first glance, but the relationships in the top and bottom row are clearly non-linear.
4. For the right column, the line of best fit is based on points sampled on another line of best fit (each pareto frontier), rather than empirical samples.
5. There are an inconsistent number of seeds used. Standard error is not reported.

---

References:

- Kumar, Aviral, et al. "Stabilizing off-policy q-learning via bootstrapping error reduction." Advances in neural information processing systems 32, 2019.
- Levine, Sergey, et al. "Offline reinforcement learning: Tutorial, review, and perspectives on open problems." arXiv preprint arXiv:2005.01643, 2020.
- Li, Qiyang, et al. "Efficient Deep Reinforcement Learning Requires Regulating Overfitting." The Eleventh International Conference on Learning Representations, 2023.
- Chen, Xinyue, et al. "Randomized Ensembled Double Q-Learning: Learning Fast Without a Model." International Conference on Learning Representation, 2021.
- D'Oro, P., Schwarzer, M., Nikishin, E., Bacon, P. L., Bellemare, M. G., & Courville, A. Sample-Efficient Reinforcement Learning by Breaking the Replay Ratio Barrier. In The Eleventh International Conference on Learning Representations, 2023.
- Haarnoja, Tuomas, et al. "Soft actor-critic: Off-policy maximum entropy deep reinforcement learning with a stochastic actor." International conference on machine learning. PMLR, 2018.
- Bhatt, A., et al. "CrossQ: Batch Normalization in Deep Reinforcement Learning for Greater Sample Efficiency and Simplicity." International Conference on Learning Representations (ICLR). 2024.

**Questions For Authors:**

I do not think the scope of an ICML rebuttal would convince me to adjust my evaluation given the current critical lack of scientific rigor in the paper.

For future revisions, the authors should aim to provide verifiable evidence of their claims.
- Transparent experimental details to make the paper reproducible.
- More complete and clearer presentation of the results, such as numerical analysis (including measures of uncertainty or standard error) and results on individual environments, and improving transparency on unknown quantities (e.g., variable "best batch size").
- Eliminate inconsistencies in evaluation protocol, removing elements such as: hand-picked values of J, inconsistent number of observations, inconsistent exponential axis scales.

----

## update after rebuttal

While I appreciate the authors attempts to improve the clarity and transparency of the results, the number of concerns I had in the original draft, on top of a number of unaddressed concerns, means I remain firm with my original score.

**Quantitative analysis:** The paper makes significant claims based only on visual inspection. The authors have not addressed this issue during the rebuttal.

**Arbitrary normalizing:** The authors use an unknown system for selecting environment scores for normalizing that may influence final results. Quote from original review below:

> This is highly exacerbated by the fact that different environments learn at very different rates. For example, for the OpenAI gym environments, the authors use SAC but only include 4 out of the 5 environments used by the original SAC paper (Table 1). Furthermore, the values of J seem arbitrarily selected. Based on the original SAC paper [Haarnoja, 2018], we can see that a score of 8500 on HalfCheetah is obtainable in fewer than 1M timesteps, but a score of 6625 on Ant would take more than 3M timesteps. Looking at more modern results, the authors of Cross-Q [Bhatt, 2024] report SAC results, where a score > 6000 is not obtained before 3M timesteps on Ant, but a score of 8500 on HalfCheetah is obtained in fewer than 1M timesteps.

**Poor clarity or missing details:** Numerous issues were detailed in the original review. For example, in Figure 2, the value of J is not described. In response, the authors mention that this value is described by a color chart in Figure 1. I do not think this is adequate organization or clarity (the reader must infer the value of J in Figure 2, by looking at Figure 1 and making a guess based on a color).

**Missing experimental analysis:** The authors show that their power law provide good fits on the single environment in Isaac gym, but do not include results for individual environments in the other domains. If the results depend on aggregation across environments, and the aggregation is based on an arbitrary normalization, then I have concerns that these results are cherry-picked. I hope to see results that show otherwise.

I hope these concerns are addressed in the next draft.

**Relation To Broader Scientific Literature:**

The contributions, if correct, provide an interesting use case for hyperparameter selection for large-scale experiments. I am not aware of any work in the RL space with similar observations, which would make this a valuable contribution as the community scales RL algorithms and applications.

**Theoretical Claims:**

N/A.

---

> ### Author Rebuttal · Authors · 2025-04-01
>
> Thank you for the detailed review and feedback! We have added several new results & clarifications to address the concerns. Specifically, for the main points:
> - We have performed the requested additional analysis to improve clarity here: https://sites.google.com/view/value-based-rl-scales/
> - We eliminated inconsistencies in evaluation protocol and report the additional results on the same link
> - To ensure transparency and reproducibility we have released the code: https://colab.research.google.com/drive/1BaqvAMb6svGojAuiOV8qFAUrZQwfPlDg?usp=sharing.
>
> We think that these changes substantially strengthen the paper and are glad that we concur on the vision of understanding how to predictably scale value-based RL. However, we also think some of the comments might stem from a misunderstanding of certain parts of the paper, and we attempt to clarify these below. We will update the paper to clarify these. Please let us know if you find your concerns addressed and if so, we would be glad if you are willing to raise your score.
>
> > what J  is
>
> As specified on Line 139, J denotes the return of the policy normalized between 0 and 1000 (with normalization detailed in Table 1).
>
> > unclear whether the result holds for all values of J
>
> It does: https://sites.google.com/view/value-based-rl-scales/home?authuser=3#h.mlu2zpb9cilb
>
> > hand-picked values of J, inconsistent number of observations
> We standardized the number of observations in this plot to be 10 and standardized the values of J to be equally spaced in log space: https://sites.google.com/view/value-based-rl-scales/home?authuser=3#h.2u2k4w2tndmu.
>
> > It is unclear how accurate the result is.
>
> We provide an additional extrapolation result where we use our model to predict the data/compute required to reach J (https://sites.google.com/view/value-based-rl-scales/home?authuser=3#h.erwyxjer0f42).
> The error is only 7.8% and 10.6% for extrapolating toward larger data and larger compute, respectively.
>
> >  y-axis follows an unknown exponential scheme
>
> Following prior work (Kaplan’20), y-axis follows a logarithmic scale. To improve readability, we also created a version that uses logarithm with base 2 (https://sites.google.com/view/value-based-rl-scales/home?authuser=3#h.8e53cf8zcw66).
>
> > unclear how the result was averaged (...), unclear whether the result holds over individual environments
>
> We shared fits across several environments to increase the generality and robustness of the results (paragraph in line 391 and Appendices B & D). On IsaacGym, we also show that it is possible to obtain a good fit on a single task.
>
> >  challenges associated with offline RL and high UTDs
>
> Following DroQ, we use LayerNorm in our SAC implementation and BRO uses LayerNorm and resets by default. This allows higher utd. We further add a new experiment with high UTD of 64: https://sites.google.com/view/value-based-rl-scales/home?authuser=3#h.mo01yp5qtkz0. Here, we use the existing hyperparameter fits based on utd 0.25...8, extrapolating hyperparameters far beyond that, and we find that the data requirement fit is still accurate.
>
> > It is unclear how accurate the relationship is between UTD and best batch size/learning rate.
>
> We provide quantitative correlations here https://sites.google.com/view/value-based-rl-scales/home?authuser=3#h.9t39ptle0tkt. We also provide quantitative evaluation of a baseline approach that uses hyperparameters optimal for UTD=2 to extrapolate to larger UTD.
>
> > It is unclear how the compute is affected by the unknown variable quantity B(σ). It is unclear how the best batch size or learning rate is determined over a collection of environments.
>
> Best choice batch size is provided in Eq (4.6). The workflow used to obtain the value is provided in section 4.3. We use different hyperparameters per environment.
>
> > It is unclear how these results [for Eq 4.5] are generated.
>
> The description the reviewer provides is correct. Please also refer to Section 4.3 for exact workflow.
>
> > How are empirical values [for Eq 4.5] averaged over a collection of environments?
>
> This is described in l864. We normalize the data requirements by per-environment median.
>
> > Figures 3 and 1 (right) do not directly show UTD.
>
> Please refer to the corresponding points in Figure 2.
>
> > The line of best fit is based on points sampled on another line of best fit (each pareto frontier), rather than empirical samples.
>
> It is not possible to use empirical samples since the grid search includes only a small number of UTDs. Estimating scaling laws, however, enables us to optimize hyperparameters with higher precision than the grid search, similar to LLM literature (Kaplan’20, Dubey’24).
>
> > There are an inconsistent number of seeds used.
>
> We used the maximum number of seeds that was feasible given our compute limitations. Different algorithms allow different number of seeds to be run in parallel.

---

> > ### Comment · Reviewer_sgc1 · 2025-04-02
> >
> > Thank you for the response.
> >
> > To the best of my knowledge, I did not misunderstand elements of the paper (at least in the cases the authors have responded to). My concern remains with the fact that many details are missing or obscured.
> >
> > While I appreciate the additional figures, I remain concerned about the lack of quantitative and statistical analysis, how the environments are averaged together (see my concerns on environment scores in point 3), and lack of transparency on key experimental quantities.
> >
> > > what J is
> >
> > To be clear, my concern is not with the definition of J, but with the fact that the value of J is not described in the Figure.
> >
> > > We provide an additional extrapolation result where we use our model to predict the data/compute required to reach J …
> >
> > I find this figure unclear. How are you measuring error? How are you computing the extrapolations?
> >
> > > We shared fits across several environments to increase the generality and robustness of the results (paragraph in line 391 and Appendices B & D).
> >
> > I find the use of isotonic regression (Appendix D) to transform the data potentially concerning. The authors justify this usage in that they can make reliable predictions (unquantified), but is the accuracy of these predictions also based on transformed data?
> >
> > > On IsaacGym, we also show that it is possible to obtain a good fit on a single task.
> >
> > The experiments with IsaacGym only use a single algorithm and environment, does this result generalize?
> >
> > > Following DroQ, …
> >
> > These details need to be in the paper.
> >
> > > Best choice batch size is provided in Eq (4.6). The workflow used to obtain the value is provided in section 4.3. We use different hyperparameters per environment.
> >
> > What is the metric used to determine the best batch size? Optimizing for compute, performance, or training error will all significantly affect final compute results. What quantities are used by each environment in the Figures?
> >
> > > This is described in l864. We normalize the data requirements by per-environment median.
> >
> > How does this translate to the small set of empirical values presented in the Figures? What is the standard error on these predictions?

---

> > > ### Author Response · Authors · 2025-04-05
> > >
> > > Thank you for the quick response and for the time to help make our paper better. We present new results. Please let us know if these responses address your concerns.
> > >
> > > - “I remain concerned about the lack of quantitative and statistical analysis”
> > >
> > > We now added confidence intervals to extrapolation results (https://sites.google.com/view/value-based-rl-scales/). Please let us know if we should add some other analysis to fully resolve your concern.
> > >
> > > - “concerns on env. scores in point 3”
> > >
> > > Could you please clarify what concerns remain from point 3? As described in our response, the difference between different environments is accounted for via normalizing.
> > >
> > > - “lack of transparency on key experimental quantities.”
> > >
> > > We have provided extensive appendices, new analysis, and released the code, with details in Appendix B. Can you clarify if there are other ways we can improve transparency: we are committed to doing so.
> > >
> > > We provide detailed responses below
> > >
> > > > What is the standard error on these predictions?
> > >
> > > We provide a new statistical analysis of the UTD extrapolation here: https://sites.google.com/view/value-based-rl-scales/home?authuser=3#h.tw8rudtiurgq. Concretely, we computed the confidence interval via bootstrapping over seeds. P-values of the hypothesis that optimal UTD doesn’t depend on budget $f$ are 0.0003, 0.000009, and 0.001 on the DMC/Gym/IsaacGym, proving that there is a dependency. The relative errors of the predictions are 1.5%, 7.7%, and 1.1%.
> > >
> > > We provide a new analysis of the Pareto frontier here: https://sites.google.com/view/value-based-rl-scales/home?authuser=3#h.erwyxjer0f42. We computed the confidence intervals via bootstrapping budget values. For extrapolating towards higher compute and higher data, the p-values of the hypothesis that data requirement doesn’t depend on UTD are 7e-12 and 8e-12. The relative errors of the predictions are 7.8%, and 10.6%.
> > >
> > > > Value of J
> > >
> > > Figure 1 contains a color bar describing the value of J. We will clarify this as shown here: https://sites.google.com/view/value-based-rl-scales/home?authuser=3#h.kajfdp9z1yj1
> > >
> > > > How are you measuring error? How are you computing the extrapolations?
> > >
> > > We reported the relative error of 7.8% and 10.6%. That is, RelativeError = (TrueValue - PredictedValue) / TrueValue. To compute it, we estimate the parameters of the fit (Eq 4.1) and then comparing predicted and held out true values.
> > >
> > > > Isotonic regression (Appendix D)
> > >
> > > Because the policy, initial states, and the environments can be stochastic, it is necessary to smooth the return curves to determine $\mathcal{D}_\textrm{J}$. However, the commonly used exponential or gaussian smoothing when reporting RL learning curves presents the need to tune the amount of smoothing. Qualitatively, isotonic regression is more reasonable and does not require a hyperparameter so it is preferred. Please refer to Figure 7 as well as this new visualization: https://sites.google.com/view/value-based-rl-scales/home?authuser=3#h.niwnmjq1h40d.
> > >
> > > Finally, we would also like to note that we do not see the use of isotonic regression by itself as a concern since this choice still allows us to fit relationships between hyperparameters, data and compute budgets that still extrapolate to producing good raw scores at scale. Hence it does not inhibit extrapolation or reliable predictions of raw scores. That said, if there are better ways to transform raw data, we can try them out. Note that since scaling laws for value-based RL are fairly unexplored, we are unaware of any standard practice.
> > >
> > > > The experiments with IsaacGym only use a single algorithm and environment, does this result generalize?
> > >
> > > We perform experiments with 3 algorithms and 3 domains. While we agree additional results would improve the paper, it is not feasible to add them due to space limitations.
> > >
> > > > These details need to be in the paper.
> > >
> > > We are not able to revise the submission on openreview. We will add the details in the final version.
> > >
> > > > What is the metric used to determine the best batch size?
> > >
> > > An insightful question! As we note on l100, since the values of return can be arbitrary, we instead select the batch size to minimize data required to reach J=800. This is the same as minimizing the number of gradient steps. As the reviewer notes, this affects the FLOPs optimality. However, we decided to report this value as we believe minimizing wall clock time is more important than FLOPs. Building a more complete model of effects of batch size on optimal FLOPs is an interesting direction of future work.
> > >
> > > > What quantities are used by each environment in the Figures?
> > >
> > > We provide the law for batch size in table 3 and specific values here: https://sites.google.com/view/value-based-rl-scales/home?authuser=3#h.3yvov5uicmra
> > >
> > > > How does this translate to the small set of empirical values presented in the Figures?
> > >
> > > In the multi-environment experiments, we fit the normalized average of the individual environments. This is described on l711.

---

### Official Review · Reviewer_beKg · 2025-03-10

**Overall Recommendation:** 4

**Summary:**

The authors demonstrate that value-based deep RL scales predictably, showing a Pareto frontier controlled by the updates-to-data (UTD) ratio. This paper shows how the optimal hyperparameters can be predicted from low-cost experiments, enabling an extrapolation to higher data or compute experiments. Validation is done using algorithms like SAC, BRO, and PQL across multiple benchmarks.

**Claims And Evidence:**

The claims are supported by a large amount of empirical evidence. The scale of the experiments lead to convincing results.

**Essential References Not Discussed:**

-

**Experimental Designs Or Analyses:**

-

**Methods And Evaluation Criteria:**

Although the evaluation criteria is fitting for the problem at hand, I would have liked to see some experiments on harder, pixel-based environments such as Atari. It would be interesting to see if these scaling laws hold with increased environment or network complexity.

**Other Comments Or Suggestions:**

I would like to see some experiments on a more complex environment such as Atari. Although I know this might seem compute intensive, the recent PQN [1] algorithm seems to be very fast.


[1] Gallici et al, 2024. Simplifying Deep Temporal Difference Learning.

**Other Strengths And Weaknesses:**

Strengths: Clear, practically helpful paper with a lot of empirical evidence.

Weaknesses: I will not claim that the experiments are limited, but, as said before, I would have liked to see some harder, pixel-based environments.

**Questions For Authors:**

-

**Relation To Broader Scientific Literature:**

The biggest improvement over broader scientific literature is that the authors also study the tradeoff between available data and compute.

**Theoretical Claims:**

The claims are empirically derived. However, the derivations seem sound.

---

> ### Author Rebuttal · Authors · 2025-04-01
>
> Thank you for the review and the positive feedback. Please let us know if the response below addresses your concerns, and if there are any concerns remaining.
>
> > Although the evaluation criteria is fitting for the problem at hand, I would have liked to see some experiments on harder, pixel-based environments such as Atari. It would be interesting to see if these scaling laws hold with increased environment or network complexity
>
> Our research vision is to understand the impact of various design choices on scaling of value-based RL algorithms. This includes a number of choices such as the choice of environments, networks, replay buffers, algorithms, etc, but unfortunately due to the limited compute budget we were left with the choice of only being able to study a few design choices only that could fit in the scope of the first paper along this direction. Therefore, we decided to first start from environments that are cheap to run (state-based) and considered different network architectures including small vanilla MLPs (used by SAC and PQL) and relatively big resnet-based models (BRO) in their specific domains.
>
> We agree that expanding to pixel-based environments would further strengthen the generalizability claim. While this is costly to do, we are happy to add preliminary results with a subset of Atari environments in the camera-ready version.
>
> In addition, in the rebuttal we have added a number of improvements suggested by other reviewers (e.g., additional quantitative analysis of proposed scaling curves, estimation of scaling laws for different levels of J), which we invite you to inspect under the following link: https://sites.google.com/view/value-based-rl-scales/ and have released the code here: https://colab.research.google.com/drive/1BaqvAMb6svGojAuiOV8qFAUrZQwfPlDg?usp=sharing. We hope that these changes increase the reviewers' confidence in our work.

---

> > ### Comment · Reviewer_beKg · 2025-04-08
> >
> > Thanks for the clarifications. I would like to see the preliminary results with a subset of Atari environments in the camera-ready version. I will keep my positive score.

---

### Official Review · Reviewer_VoCp · 2025-03-12

**Overall Recommendation:** 4

**Summary:**

This paper investigates the scalability and predictability of value-based RL using TD learning. It establishes predictable, hypothetic equations between three key hyperparameters (batch size, learning rate, and UTD ratio) and shows that data and compute requirements for a given performance lie on a Pareto frontier. By modeling the tradeoff between data and compute, the authors predict resource needs and optimal hyperparameter settings for large-scale experiments based on low-budget data. Finally, they empirically demonstrate that such findings can be extended to algorithms like SAC, BRO, and PQL and domains such as DeepMind Control, OpenAI Gym, and IsaacGym.

**Claims And Evidence:**

The authors make several key claims regarding the predictability of scaling in value-based RL, which are well-supported, as the authors provide empirical fits, validate their model across different datasets, and show that extends to well-estabilished algorithms like SAC, BRO, and PQL.

**Essential References Not Discussed:**

I think most references in mind were present. However, more works on scaling model size [1] and UTD ratio [2, 3] in RL could be included:

1. SimBa: Simplicity Bias for Scaling Up Parameters in Deep Reinforcement Learning., ICLR'25.
2. DrM: Mastering Visual Reinforcement Learning through Dormant Ratio Minimization., ICLR'24.
3. The Dormant Neuron Phenomenon in Deep Reinforcement Learning., ICML'23

**Experimental Designs Or Analyses:**

The experimental setup is generally well-structured, but a key concern is the limited selection of baseline algorithms. The rationale for choosing SAC, BRO, and PQL is weak and not fully justified. For example, I don't think BRO is the state-of-the-art algorithms in Gym; SimBa [1] offers a more practical and performant alternative by simply modifying SAC’s architecture, which could help assess the impact of network design on the proposed predictability framework. Additionally, the paper does not compare against other well-established state-of-the-art methods such as MR.Q [2]. I believe resolving my concern would enhance the evaluation and strengthen the study’s claims.

[1] SimBa: Simplicity Bias for Scaling Up Parameters in Deep Reinforcement Learning., ICLR'25.

[2] Towards General-Purpose Model-Free Reinforcement Learning., ICLR'25.

**Methods And Evaluation Criteria:**

The method for proposing their claim and empirical relationships are well-constructed and appropriate for the problem. Their study clearly formulates and supports the predictability of value-based RL scaling.

**Other Comments Or Suggestions:**

n/a

**Other Strengths And Weaknesses:**

n/a

**Questions For Authors:**

1. The study establishes predictable scaling laws for batch size, learning rate, and the UTD ratio, which are well-supported by empirical evidence. However, other parameters such as model size, weight decay, and target network update rates are also known to influence training stability and performance in deep RL. I wonder if the authors have considered these configurations too.
2. It was surprising that the compute-data pareto frontiers (Figure 1, left) for OpenAI Gym seems to be much flatter than other environments. Do the authors have any intuition about this phenomenon?

**Relation To Broader Scientific Literature:**

This paper extends scaling law research from supervised learning to value-based reinforcement learning (RL), providing a contribution towards a large-scale RL foundation model.

**Theoretical Claims:**

No theoretical claims were made.

---

> ### Author Rebuttal · Authors · 2025-04-01
>
> Thank you for valuable feedback regarding our work. In accordance with your suggestions, we have made a number of changes to our manuscript. These include expanded discussion of limitations and experimental design. We discuss these in detail below:
>
> > (other parameters such as model size, weight decay, and target network update rates are also known to influence training stability and performance in deep RL
>
> While studying a number of these factors is definitely along our vision of better understanding the behavior of scaling value-based RL algorithms along various axes, we were faced with a decision of only studying certain axes for the submission due to a limited compute budget. Therefore, we chose to initiate our study into this line of work with the most fundamental hyperparameters such as batch size, learning rate, and UTD values. We are now working to extend this line of work to study the impact of model size and target network update rates as the next piece of work along this research vision. We will also add the following discussion to limitations of this paper:
>
> _Whereas this study investigated the relationship between learning rates, batch sizes, and UTDs, there are a variety of other design choices and hyperparameters that could potentially impact the scaling profile of a given algorithm. In particular, previous works have shown that model size, weight decay, and target network update strategy can significantly affect the training dynamics between different UTD values._
>
> > It was surprising that the compute-data pareto frontiers (Figure 1, left) for OpenAI Gym seems to be much flatter than other environments. Do the authors have any intuition about this phenomenon?
>
> Indeed, it seems that the effectiveness of UTD is benchmark-specific, where increased UTD leads to pronounced sample efficiency improvements in some environments, and substantially less pronounced in others. We believe that studying the underlying phenomena that lead to such differences is an exciting avenue for future research.
>
> >  (...) The rationale for choosing SAC, BRO, and PQL is weak and not fully justified. SimBa [1] offers a more practical and performant alternative…
>
> While in general, one can study scaling laws for any value-based RL algorithm, we were faced with the tough choice of only studying a few algorithms due to limited compute. We chose SAC, BRO, and PQL because prior work has shown them to be effective in the respective benchmarks (Gym, DMC, Isaac Gym) under different UTD configurations. We will add this justification to the paper.
>
> Our framework is independent and can be applied to both BRO and SimBa, which are both methods focused on scaling the number of parameters. We observed consistent findings across a wide class of SAC-derived methods and therefore believe the results would also extend to SimBa. Due to compute constraints, we were unable to run SimBa experiments for rebuttal.
>
> We also note that our goal is not to argue that our scaling laws give rise to the best possible value-based RL algorithm or that we are able to attain the best possible results in our experiments, but rather our point is to simply show that we can predict the behavior of value-based RL in larger-scale settings using small-scale experiments. We believe that showing this for a certain class of algorithms is still valuable as a starting step in this direction of research. We believe future state of the art methods will use our framework to improve performance, closing the loop.
>
> > Additionally, the paper does not compare against other well-established state-of-the-art methods such as MR.Q
>
> Thank you for pointing us towards that work, which we now cite. We agree that studying the scaling capabilities of this general algorithm would be particularly interesting, however we were not able to include MrQ in our submission as it was arXived after the ICML deadline.
>
> > However, more works on scaling model size [1] and UTD ratio [2, 3] in RL could be included:
>
> Thank you for pointing these missing references, we added all of them to our manuscript.
>
> Beyond the changes described above, we add a number of improvements suggested by other reviewers (e.g., additional quantitative analysis of proposed scaling curves, estimation of scaling laws for different levels of J), which we invite you to inspect under the following link: https://sites.google.com/view/value-based-rl-scales/ and have released the code here: https://colab.research.google.com/drive/1BaqvAMb6svGojAuiOV8qFAUrZQwfPlDg?usp=sharing. We hope that these changes increase the reviewers' confidence in our work. If so, we kindly ask to consider updating the score of our work.

---

> > ### Comment · Reviewer_VoCp · 2025-04-02
> >
> > Thank you for the clarification!
> > I have updated my score from 3 to 4. I really enjoyed reading this paper and would be happy to see it discussed in the community.
> >
> > One particularly interesting point—highlighted in Figure 5—is that small batch sizes tend to perform well in high UTD regimes. While this may be known to some practitioners, it is not widely recognized, and the paper presents it. I recommend citing the following relevant work:
> > - Small Batch Deep Reinforcement Learning, Ceron et al., NeurIPS 2023
> >
> > A minor suggestion regarding Section 4.2: the current framing suggests that batch size relates to overfitting and learning rate to plasticity. However, these two factors are highly intertwined, and the relationship is not strictly one-to-one. It may be clearer to frame this section as exploring how both batch size and learning rate jointly affect plasticity and overfitting, rather than assigning each concept to a single factor. This would help avoid potential misconceptions, especially for readers less familiar with the literature.

---

> > > ### Author Response · Authors · 2025-04-07
> > >
> > > We will update 4.2 accordingly and add "Small Batch Deep Reinforcement Learning" to our discussion for the camera-ready version.
> > >
> > > Thank you for updating the score and for helping to improve the quality of our manuscript.

---

### Decision · Program_Chairs · 2025-05-01

**Decision:**

Accept (poster)

**Comment:**

The paper examines the scalability and predictability of value-based RL through extensive empirical experiments.

During the review stage, the paper received conflicting comments. While reviewers concur that the problem is significant and the observations and results presented in the paper are valuable, the primary concern lies in transparency and reproducibility. Some of the experiments appear to be cherry - picked. I suggest that the authors include all experiments in the revised version and conduct a failure case study. In my opinion, a failure case study does not diminish the value of a research paper; rather, it enhances it. Overall, I would recommend accepting the paper if the authors can take these suggestions into account.